# BCL7A and BCL7B potentiate SWI/SNF-complex-mediated chromatin accessibility to regulate gene expression and vegetative phase transition in plants

Switch defective/sucrose non-fermentable (SWI/SNF) chromatin remodeling complexes are multi-subunit machineries that establish and maintain chromatin accessibility and gene expression by regulating chromatin structure. However, how the remodeling activities of SWI/SNF complexes are regulated in eukaryotes remains elusive. B-cell lymphoma/leukemia protein 7 A/B/C (BCL7A/B/C) have been reported as subunits of SWI/SNF complexes for decades in animals and recently in plants; however, the role of BCL7 subunits in SWI/SNF function remains undefined. Here, we identify a unique role for plant BCL7A and BCL7B homologous subunits in potentiating the genome-wide chromatin remodeling activities of SWI/SNF complexes in plants. BCL7A/B require the catalytic ATPase BRAHMA (BRM) to assemble with the signature subunits of the BRM-Associated SWI/SNF complexes (BAS) and for genomic binding at a subset of target genes. Loss of BCL7A and BCL7B diminishes BAS-mediated genome-wide chromatin accessibility without changing the stability and genomic targeting of the BAS complex, highlighting the specialized role of BCL7A/B in regulating remodeling activity. We further show that BCL7A/B fine-tune the remodeling activity of BAS complexes to generate accessible chromatin at the juvenility resetting region (JRR) of the microRNAs MIR156A/C for plant juvenile identity maintenance. In summary, our work uncovers the function of previously elusive SWI/SNF subunits in multicellular eukaryotes and provides insights into the mechanisms whereby plants memorize the juvenile identity through SWI/SNF-mediated control of chromatin accessibility.

The dynamic control of chromatin accessibility is critical for establishing and maintaining timely and precise gene expression programs[1]. Switch defective/sucrose non-fermentable (SWI/SNF) complexes are evolutionarily conserved molecular machines that regulate chromatin architecture to modulate genomic accessibility via sliding or ejection of nucleosomes in an ATP-hydrolysis-dependent manner[2–6]. These complexes have been found to serve essential roles in regulating developmental reprogramming and genomic processes such as transcription in yeast, drosophila, mammals, and plants[4,7,8]. The chromatin remodeling activities of SWI/SNF are continuously required to control genome accessibility and, therefore, must be tightly regulated to ensure fidelity in genomic processes[2,3,9]. Dysregulation of SWI/SNF

✉ e-mail: lichlong3@mail.sysu.edu.cn

activities leads to developmental disorders often associated with aberrant gene transcription programs. However, despite the conservation and biological relevance of the SWI/SNF complexes, how their remodeling activities at target genes are regulated is largely unknown, presenting a fundamental gap in our understanding of SWI/SNF function in eukaryotes.

In humans, SWI/SNF complexes (also termed BAF) can be divided into three distinct subtypes: canonical BAF (cBAF), polybromo-associated BAF (PBAF), and non-canonical BAF (ncBAF)[10,11]. They share a set of core subunits, such as BRG1, SMARCD, ACTL6A/B, and BCL7A–BCL7C, but are distinguished by the inclusion of subtype-specific ones: i.e., the cBAF by DPF1, DPF2, ARID1A, and ARID1B; the PBAF by PBRM1, PHF10, ARID2, and BRD7; and the ncBAF by BRD9, GLTSCR1, and GLTSCR1L[10,11]. In the flowering plant Arabidopsis (*Arabidopsis thaliana*), four SWI2/SNF2 ATPases: BRAHMA (BRM), SPLAYED (SYD), MINUSCULE1 (MINU1), and MINU2 have been identified[8,12,13]. Recently, we and others showed the existence of three concurrently expressed plant SWI/SNF family sub-complexes that have specific subunits and are separately assembled on a mutually exclusive catalytic subunit, BRM, SYD, or MINUs. These three plant SWI/SNF sub-complexes were termed BAS (BRM-Associated SWI/SNF complex), SAS (SYD-Associated SWI/SNF complex), and MAS (MINU-Associated SWI/SNF complex), respectively[14–17].

Dissection of the subunit-specific contributions to SWI/SNF complex function is needed to advance our mechanistic understanding of the role played by SWI/SNF complexes in eukaryotes. Recent studies in humans and plants have begun to reveal the distinct roles of individual subunits in regulating the assembly/stability and genomic targeting of the SWI/SNF complexes. For instance, in humans, SMARCE1, ARID1A/B, ARID2, SMARCC, SMARCD, and BRD9 have been shown to be required for the assembly and stability of the complexes[10,11,18,19], whereas the SMARCB1 and BRD9 subunits are responsible for the regulation of the genomic targeting of the complexes[10,20,21]. In plants, we recently showed that BRIP1 and BRIP2, homologs of the human ncBAF subunits GLTSCR1/1 L, are required for the assembly of Arabidopsis BAS SWI/SNF complexes[14]. In addition, we found that three bromodomain-containing subunits, BRD1, BRD2, and BRD13, play an essential role in facilitating the targeting of BRM-containing SWI/SNF complexes to chromatin in plants[15]. Interestingly, depletion of the above-studied subunits results in compromised remodeling activities of SWI/SNF complexes at target genes in both humans and plants, but this reduction is a secondary consequence of impaired complex assembly or genomic targeting following the loss of those subunits. So far, the subunit(s) that solely regulate the chromatin remodeling activities of SWI/SNF complexes without affecting complex assembly and/or genomic targeting has not been defined in any eukaryotes.

Multicellular eukaryotes undergo multiple developmental phase transitions during their life cycles. The correct timing of these phase transitions is critical to survival and reproductive success. Plant vegetative development can be divided into the juvenile phase and the adult phase[22,23]. The juvenile-to-adult phase transition (also termed the vegetative phase transition) is triggered by a progressive decrease in the level of miR156, a microRNA that is conserved throughout the plant kingdom[22,24–26]. miR156 accumulates at high levels in juvenile seedlings before the vegetative phase transition and is necessary for maintaining juvenile identity[24,27–29]. Remarkably, a recent study showed that *MIR156A* and *MIR156C* (*MIR156A/C*), the two major contributors to miR156 levels, are re-activated de novo during embryogenesis and that the DNA accessibility level of an upstream region near the *MIR156A/C* transcription start sites (TSS) is positively correlated with their transcriptional activity[30]. This region, termed the juvenility resetting region (JRR), was found to be accessible in both embryos and juvenile seedlings where *MIR156A/C* are highly expressed but becomes inaccessible in the adult phase when *MIR156A/C* are silenced. Accordingly, the JRR is

required for the de novo activation of *MIR156A/C* in embryos and for plant juvenility[30]. However, the molecular mechanisms that engage the chromatin accessibility at the JRR of *MIR156A/C* for juvenile identity maintenance remain largely unclear. The B3 DNA-binding domain transcription factor LEAFY COTYLEDON2 (LEC2) was reported to be required for enhanced chromatin accessibility at *MIR156A/C* in embryos, but the mechanism was unknown[30]. Moreover, because *LEC2* does not express in vegetative tissues, how this accessible chromatin environment generated by LEC2 in seeds is sustained in juvenile seedlings is obscure.

Recently, several studies identified two homologous proteins of unknown function as Arabidopsis pan-SWI/SNF subunits[14–17,31,32], which are distant orthologs of the human B-cell lymphoma/leukemia protein 7 A/B/C (BCL7A/B/C). The Arabidopsis BCL7A and BCL7B were also named as BCL-domain homolog 1 (BDH1) and BDH2[33]. The *bcl7a bcl7b* double mutant Arabidopsis exhibits pleiotropic phenotypes, implying aberrant SWI/SNF function in the absence of BCL7A/B in plants[15]. BCL7A/B/C were also shown to be members of human SWI/SNF complexes and frequently undergo biallelic inactivation in diffuse large B-cell lymphomas[11,34,35]. A number of clinical studies have reported that the BCL7 family is involved in cancer incidence, progression, and development, and functions as tumor suppressors[35,36]. Although BCL7s have been reported as subunits of SWI/SNF complexes for decades in animals and recently in plants, the role of BCL7 subunits in SWI/SNF complexes remains unknown.

In this study, we demonstrate that Arabidopsis BCL7A and BCL7B are required for potentiating SWI/SNF-mediated genome-wide chromatin accessibility. BCL7A/B require BRM to incorporate into the BAS complexes and for their genomic binding at a subset of target genes. Depletion of BCL7A/B results in a retained ability of BAS complexes to assemble and to bind to target chromatin but an inability to create DNA accessibility to the levels of wild-type (WT) complexes. Likewise, BCL7A/B are also responsible for the SAS- and MAS-dependent chromatin accessibility regulation. Finally, we demonstrate that BCL7A/B and BRM act interdependently to generate accessible chromatin at the JRR in *MIR156A/C*, thereby activating the expression of *MIR156A/C* in juvenile seedlings to prevent the precocious acquisition of adult identity. Together, our results uncover the unique function of the previously elusive subunits in fine-tuning the chromatin accessibility regulation ability of SWI/SNF complexes, thus decoupling the assembly and genomic binding of the complexes from their nucleosome remodeling activity. Our results also advance mechanistic understanding of how plants prevent precocious juvenile phase transition.

## Results
### BCL7A and BCL7B interact and co-localize with BRM on chromatin

To define the molecular mechanisms underlying the assembly, genomic targeting, and activities of SWI/SNF complexes, we previously performed an immunoprecipitation-mass spectrometry-based analysis (IP-MS) using a stable transgenic Arabidopsis that expressed green fluorescent protein (GFP)-tagged BRM driven by its native promoter in the *brm-1* null mutant background[14,15]. This analysis allowed us to identify a plant BRM-containing SWI/SNF complex which was recently named as the BAS complexes[14–17]. In addition to the BRIP1/2 and BRD1/2/13 subunits[14,15], we isolated two homologous proteins of unknown function, encoded by *AT4G22320* and *AT5G55210*, respectively (Supplementary Fig. 1a)[17]. The two proteins (named as BCL7A and BCL7B) contained 238 and 168 amino acids, respectively, and showed sequence similarity mainly to the N-terminal region of the mammalian BCL7A/B/C proteins (Supplementary Fig. 2).

To further confirm that BCL7A/B are members of BAS complexes, we performed several independent assays. Bimolecular fluorescence complementation (BiFC) analyses using *Nicotiana benthamiana* leaves detected positive fluorescent signals in nuclei when co-expressing N-

terminal YFP-fused BCL7A/B and C-terminal YFP-tagged BRM. By contrast, no signal was observed when an unrelated nucleus-localized protein HAT3 (HOMEOBOX-LEUCINE ZIPPER PROTEIN 3) was used (Supplementary Fig. 1b). Moreover, we performed co-immunoprecipitation (Co-IP) using a *Nicotiana benthamiana* transient expression system and observed that BCL7A or BCL7B interacted with the N-terminal part of BRM (amino acids 1–952) (Fig. 1a, b). FLAG-tagged BCL7A or BCL7B also immunoprecipitated hemagglutinin (HA)-tagged BRIP2, BRD2, and SWI3C (Fig. 1c, d), which are signature subunits of the BAS SWI/SNF complexes[14,15].

Previous yeast two-hybrid assays failed to detect a direct interaction of BCL7A/B with any subunits of the SWI/SNF complexes[16]. We used a protein pull-down assay to test whether bacterially expressed glutathione *S*-transferase (GST)-BCL7A/B interacted with BRM and found that GST-BCL7A and -BCL7B, but not GST alone, pulled down BRM at the 422–952 amino acids that contain the QLQ and HSA domains (Fig. 1e), suggesting the direct physical interaction between BCL7A/B with BRM. Taken together, these results provide evidence that BCL7A and BCL7B directly interact with BRM and are components of the BAS SWI/SNF complexes.

We next compared the genome-wide localizations of BCL7A and BCL7B with the localization of BRM through chromatin immunoprecipitation followed by high-throughput sequencing (ChIP-seq) using Arabidopsis transgenic lines expressing GFP-tagged BCL7A or BCL7B under the control of their respective native promoters in the corresponding single mutant backgrounds (*pBCL7A:BCL7A-GFP bcl7a* and *pBCL7B:BCL7B-GFP bcl7b*)[17]. Consistent with our recent observations[17], correlation analysis showed correlated BCL7A and BCL7B co-localization with BAS complexes subunits (Supplementary Fig. 3a). We also observed a high correlation for BCL7A or BCL7B co-localization with BRD1/2/13-recognized H4K5ac ($r = 0.67$)[15,37] but not with repressive H3K27me3 (Supplementary Fig. 3a). For comparison, the correlation between BRM and H4K5ac was in a similar range ($r = 0.62$) (Supplementary Fig. 3a). Consistently, heatmap analysis showed that BCL7A, BCL7B, BRIP1/2, BRD1/2/13, and H4K5ac but not H3K27me3, all exhibited substantial enrichments at BRM binding peak summits (Fig. 1f). When we repeated the co-occupancy analysis using enrichment relative to TSSs, rather than peak summits, we observed that BRM and H4K5ac, but not H3K27me3, were enriched at BCL7A- and BCL7B-occupied genomic regions (Supplementary Fig. 3b). We further compared BCL7A-bound genes ($n = 10,586$) with randomly selected BCL7A-unbound genes ($n = 10,586$), finding a significant enrichment of BRM or H4K5ac at BCL7A-bound versus BCL7A-unbound genes (Fig. 1g). By contrast, BCL7A-bound genes were depleted of H3K27me3 (Fig. 1g). Likewise, BRM and H4K5ac were significantly enriched at BCL7B-bound compared with at BCL7B-unbound genes (Supplementary Fig. 3c). ChIP-seq data showing the co-enrichment of BCL7A and BCL7B with BRM and H4K5ac, but not with H3K27me3, at selected loci were presented in Fig. 1h, and were validated by independent ChIP-qPCR (Supplementary Fig. 3d). Taken together, these results indicate that BCL7A/B show a genome-wide co-localization with BRM on chromatin and that all three proteins preferentially bind to active gene loci.

We also compared SYD- and MINU2-binding genes[17] with our BCL7A/B binding profile. The binding of BCL7A and BCL7B was significantly overlapped with those of SYD and MINU2 (Supplementary Fig. 4a, b), consistent with the notion that BCL7A/B are pan-SWI/SNF components[16,17].

## BCL7A/B require BRM for assembling with the signature subunits of the BAS complexes

We next sought to evaluate whether the incorporation of BCL7A/B into the BAS complexes was dependent on BRM. To this end, we introduced the loss-of-function *brm-1* mutation into the *BCL7A-GFP* or *BCL7B-GFP* transgenic line (*pBCL7A:BCL7A-GFP bcl7a brm-1* and

*pBCL7B:BCL7B-GFP bcl7b brm-1*) and preformed IP-MS analyses. In the *brm-1* null mutants, the protein levels of BCL7A and BCL7B were largely unaffected (Fig. 2a–e). The BAS-complex signature subunits, including SWI3C, BRIP2, and BRD1/2/13, were reduced but not disappeared in the *brm-1* mutants[17]. However, in the absence of BRM, neither BCL7A nor BCL7B was able to immunoprecipitate SWI3C, BRIP2, and BRD1/2/13 (Fig. 2a–c, Supplementary Fig. 5a–d, Supplementary Data 1). Hence, these data imply that BCL7A/B require BRM for incorporating/assembling with the signature subunits (SWI3C, BRIP2, and BRD1/2/13) to form the BAS complexes. Notably, SWI3A and SWI3D were immunoprecipitated by BCL7A or BCL7B in the *brm-1* mutant backgrounds with peptide numbers comparable to WT background. Thus, BRM is not necessary for BCL7A/B to immunoprecipitate SWI3D and SWI3A, in line with previous reports showing that BRM does not form complexes with SWI3D and SWI3A[16,17].

Interestingly, we observed a higher number of SWI3D peptides immunoprecipitated by BCL7B-GFP compared to SWIC (Fig. 2a, b). A similar result was also observed in BCL7A-GFP samples (Supplementary Fig. 5a–c and Supplementary Data 1). These results imply the possibility that BCL7A and BCL7B are more frequently associated with SWI/SNF complexes containing SWI3D than those containing SWI3C. Moreover, SYD and MINU1/2 ATPases were identified in BCL7A/B-GFP samples (Fig. 2a–c and Supplementary Fig. 5a–c). Notably, SYD showed relatively higher peptide numbers compared to BRM in BCL7A-GFP samples (Supplementary Fig. 5a–c). However, the scenario reversed in BCL7B-GFP plants, where BRM was identified with a higher number of peptides compared to SYD and MINU1/2 (Fig. 2a–c). This observation suggests that BCL7A might prefer to interact with SYD, whereas BCL7B exhibits a higher affinity for BRM ATPase compared to SYD and MINU1/2.

We further determined the impact of BRM loss on the genome-wide targeting of BCL7A and BCL7B through ChIP-seq. Notably, and in line with their absence from the BAS complex upon BRM loss, we observed a marked attenuation in BCL7A/B targeting on the genome (Fig. 2f–m). Indeed, the occupancy levels of BCL7A- and BCL7B-GFP in *brm-1* were quantitatively reduced at 2,079 and 9,499 sites typically occupied by BCL7A and BCL7B in the WT, respectively, while they were increased at much fewer sites (Fig. 2f, g and Supplementary Fig. 5e). Consistently, the average occupancy intensity of BCL7A- or BCL7B-GFP at those sites was also substantially decreased upon the depletion of BRM (Fig. 2h–k). At the single-gene level, ChIP-seq signals for BCL7A and BCL7B on individual loci were largely diminished in the *brm-1* mutant background (Fig. 2l), and these results were independently validated by ChIP-qPCR analysis (Fig. 2m). Moreover, we found that the genomic sites where BCL7A/B were reduced in *brm-1* showed a significant overlap with BRM and BCL7A/B co-target sites, indicating that the localization of BCL7A/B on many target genes is dependent on the direct presence of BRM ATPase (Supplementary Fig. 6a, b). Notably, the BCL7s binding sites that were unaffected in *brm-1* were significantly enriched in both SYD and MINUs binding sites, implying that the binding of BCLs at those sites may be directly regulated by SYD and/or MINUs (Supplementary Fig. 6c–f). Consistently, SYD and MINU1/2 formed complexes with BCL7A/B (Fig. 2a–c and Supplementary Fig. 5a–c)[16,17,33].

## Loss of BCL7A/B does not affect BAS complex stability and genomic targeting

Our data so far suggested that BCL7A/B can be incorporated into the BAS complexes and recruited to many target genes by BRM; however, as with their orthologs in animals, the functions of BCL7A/B in SWI/SNF complexes are unknown. We first evaluated the contribution of BCL7A/B to the integrity and subunit stability of the complexes. To this end, we introduced the BRM-GFP transgene into the *bcl7a bcl7b* double mutant (*pBRM:BRM-GFP brm-1 bcl7a bcl7b*). Compared with the WT, the *BCL7A/B* mutations did not substantially

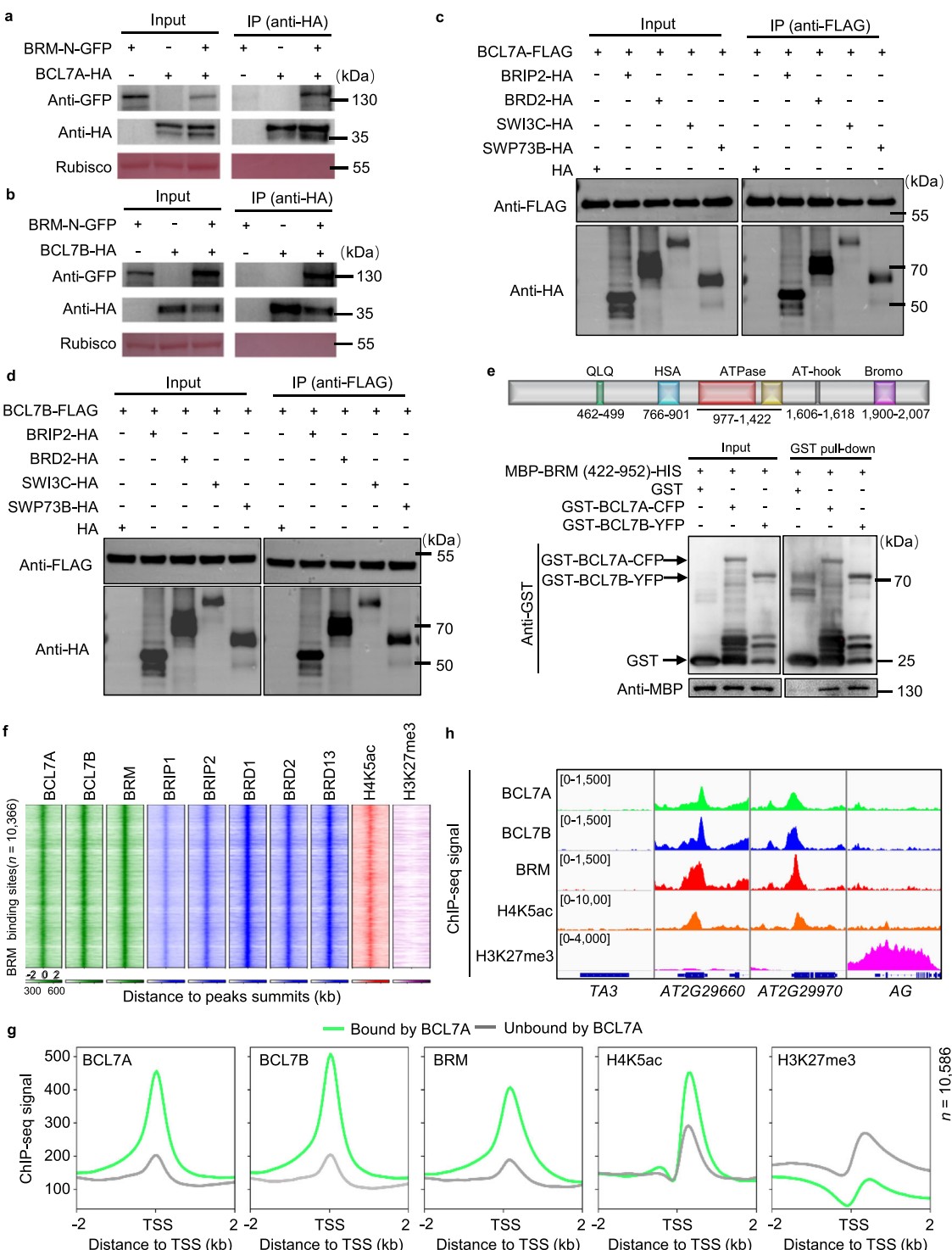

**Fig. 1 | BCL7A and BCL7B interact and co-localize with BRM on chromatin.** Co-immunoprecipitation showing the interaction of BCL7A (**a**) or BCL7B (**b**) with BRM-N terminal (1–952 amino acids). BRM-N-GFP was co-immunoprecipitated with anti-HA-agarose beads from *Nicotiana benthamiana* leaves that co-expressed BRM-N-GFP and BCL7A/B-HA. Ponceau S staining of Rubisco was shown as the loading control. Co-immunoprecipitation showing that BCL7A (**c**) or BCL7B (**d**) interacts with *Arabidopsis* SWI/SNF complexes core subunits. **e** Pull-down assays displaying that BCL7A and BCL7B interact with BRM (422-952 amino acids). The schematic illustration of the BRM conserved domains was shown on top. Data in (**a**, **b**, **c**, **d** and **e**) represent at least 3 biological experiments. **f** Heat map representations of ChIP-seq of BRM, BCL7A/B, BRIP1/2, BRD1/2/13, and histone modifications (H4K5ac and

H3K27me3). The rank order is from highest to lowest according to the BRM binding signal. log$_2$ enrichment was normalized to reads per genome coverage. Read counts per gene were averaged in 50-nucleotide (nt) bins. **g** Average distribution of BCL7A, BCL7B, BRM, and histone modifications (H4K5ac and H3K27me3) ChIP-seq reads at BCL7A-bound genes versus BCL7A-unbound genes. Read counts per gene were summed in 50-nt bins. **h** IGV views of ChIP-seq signals of BCL7A, BCL7B, BRM, H4K5ac, and H3K27me3 at representative genes. The blue diagrams underneath indicate gene structure. The y-axis scales represent shifted merged MACS2 tag counts for every 10 bp window. *TA3* and *AGAMOUS* were included as negative control loci. Source data are provided as a Source Data file.

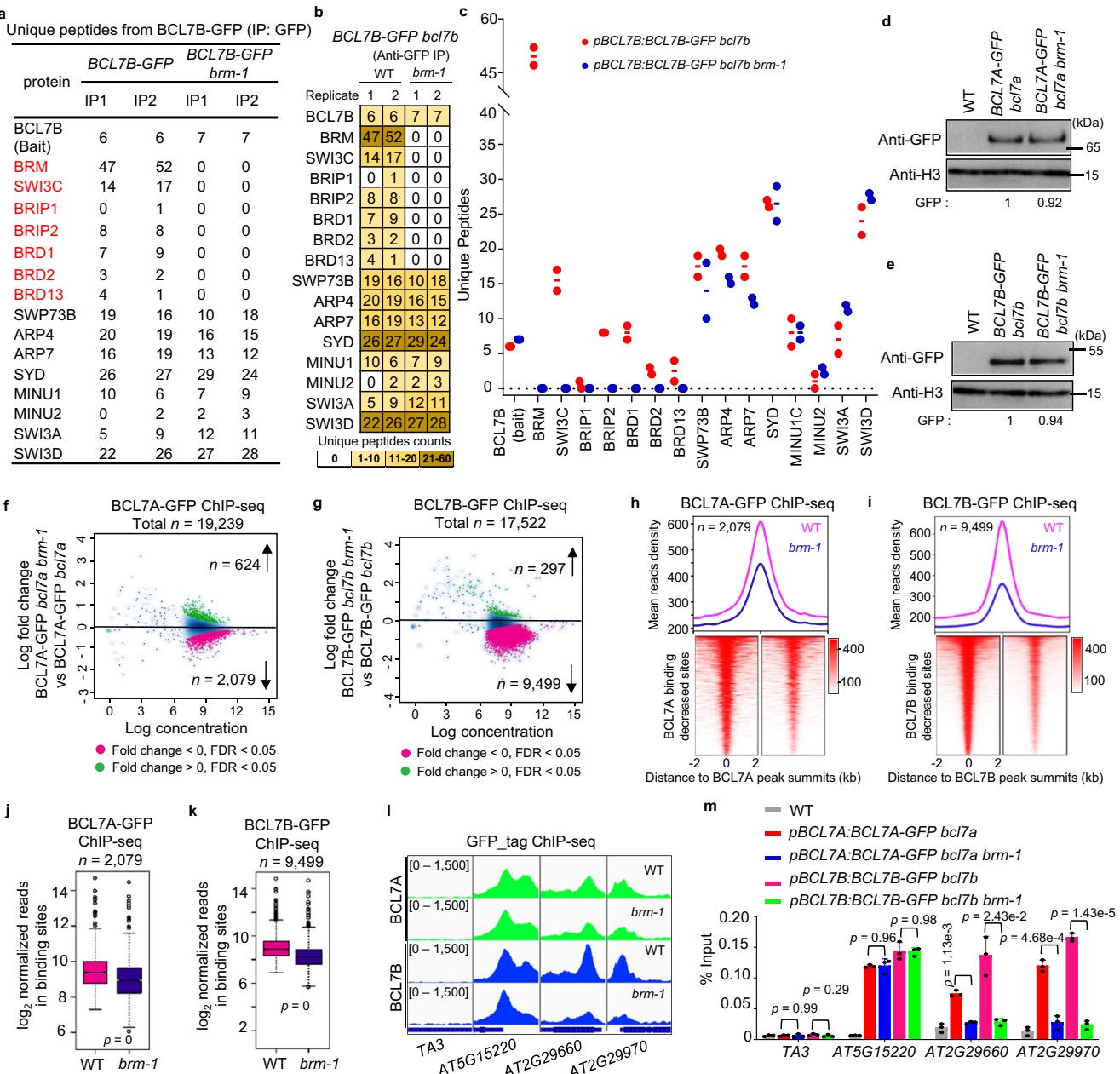

**Fig. 2 | BCL7A/B require BRM for assembling with the signature subunits of the BAS-complexes. a–c** Unique peptide numbers of BRM-SWI/SNF complex subunits identified by IP-MS in *BCL7B-GFP bcl7b* and *BCL7B-GFP bcl7b brm-1*. BAS-specific subunits are in red. Note: SYD, MINU1, MINU2, SWI3A, and SWI3D are not subunits of BAS complexes. Error bars are presented as mean ± s.d. from two biological replicates. Immunoblots showing the relative protein levels of BCL7A-GFP (**d**) and BCL7B-GFP (**e**) in the WT and *brm-1* background. The numbers at the bottom represent amounts normalized to the loading control, histone H3. WT is used as a GFP-free control. Data in (**d, e**) represent at least 3 biological experiments. Fold change (log₂) in BCL7A (**f**) or BCL7B (**g**) occupancy between WT and *brm-1* background. Differentially occupied sites with a false discovery rate (FDR) < 0.05 are highlighted in purple (decreased sites) and green (increased sites). At the top, metagene plot representations of the mean BCL7A (**h**) or BCL7B (**i**) occupancy at

regions that showed decreased BCL7A or BCL7B binding in *brm-1*. At the bottom, heatmap showing the occupancy signal of BCL7A (**h**) or BCL7B (**i**) at regions that showed decreased BCL7A or BCL7B binding in *brm-1*. Box plots showing read counts at regions showed decreased BCL7A (**j**) or BCL7B (**k**) binding in *brm-1* compared with WT. In box plots, center line and bounds of box represent median value and the interquartile range (IQR), respectively. Whiskers extend within 1.5 times the IQR. Hollow circles represent outlier values that are outside the range of the whiskers. *P* values were calculated with the two-tailed Student's *t* test. **l** IGV views of ChIP-seq signals at representative genes. **m** Validation of ChIP-seq signals at representative loci by ChIP-qPCR. *TA3* serves as a negative control. *AT5G15220* that showed maintained binding signal serves as the control locus. Error bars are presented as mean ± s.d. from three biological replicates. *P* values were calculated with the two-tailed Student's *t* test. Source data are provided as a Source Data file.

decrease the BRM-GFP mRNA and protein levels (Fig. 3a and Supplementary Fig. 7a). Furthermore, immunoprecipitation of BRM-GFP followed by mass spectrometry analysis showed that mutations of *BCL7A/B* did not result in a substantial decrease in peptide numbers of BAS-complex subunits, including SWI3C, BRIP2, BRD1, SWP73B, and ARP4/7 (Fig. 3b and Supplementary Fig. 7b). To confirm this, we

introduced a *SWI3C-FLAG* transgene into the *BRM-GFP* line in the WT and *bcl7a bcl7b* double mutant backgrounds and again did not observe changes in either total or BRM-bound SWI3C protein levels upon loss of BCL7A/B (Fig. 3c). Collectively, these results indicate that the assembly of the BAS SWI/SNF complexes is not dependent on BCL7A/B.

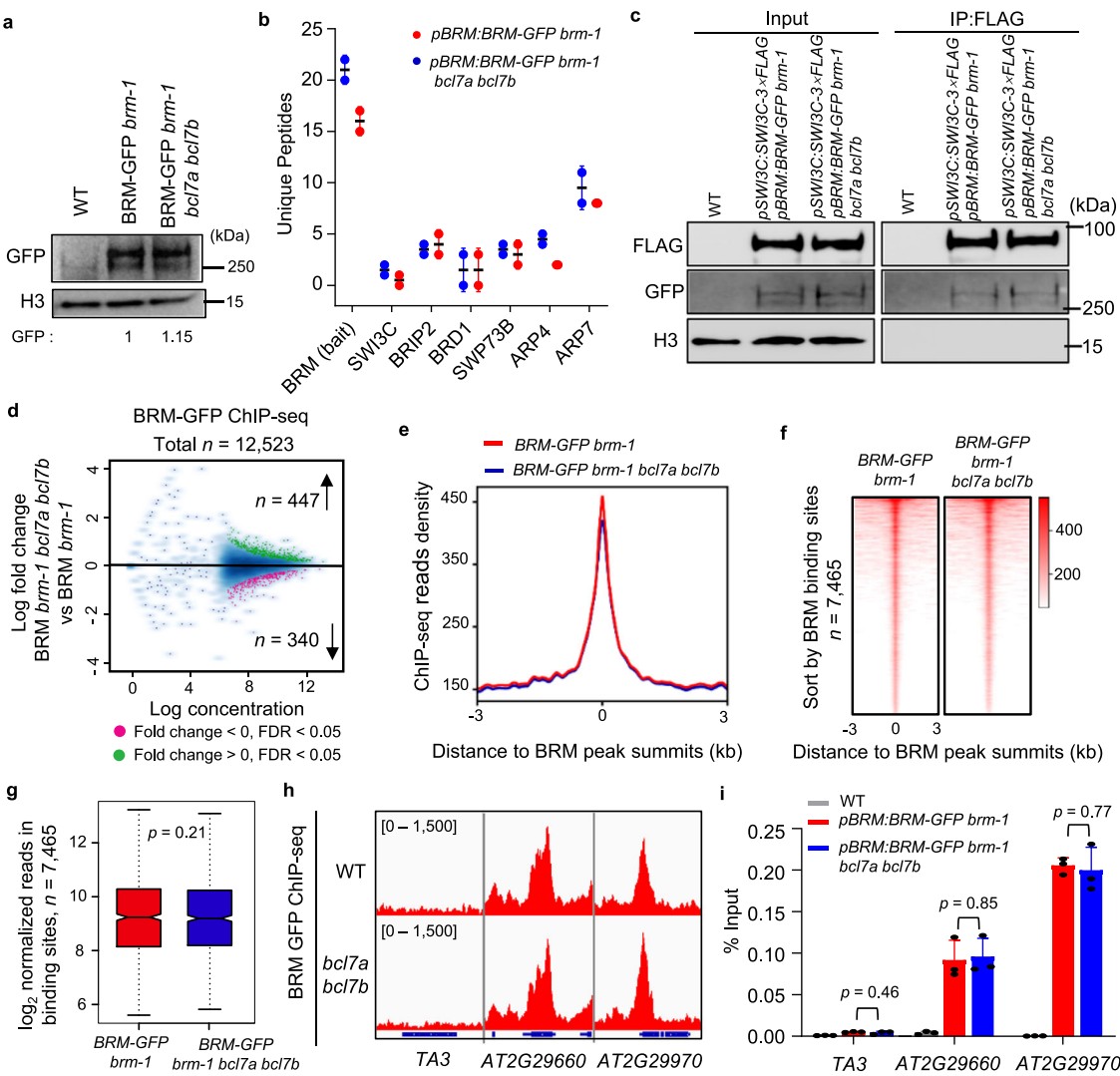

**Fig. 3 | Loss of BCL7A/B does not affect BRM-SWI/SNF complexes stability and genomic targeting. a** Immunoblot analysis showing the relative protein levels of BRM-GFP in WT and *bcl7a bcl7b* background. The numbers at the bottom represent amounts normalized to the loading control, histone H3. WT is used as a GFP-free control. **b** Mass spectrometry analysis showing numbers of peptides corresponding to SWI/SNF complex subunits recovered by immunoprecipitation of BRM-GFP in the WT and *bcl7a bcl7b* background. Error bars are presented as mean ± s.d. from two biological replicates. **c** Immunoblot showing the levels of SWI3C-3×FLAG and BRM-GFP from co-IP experiments with anti-FLAG antibody in the genetic backgrounds indicated above lanes. For each plot, the antibody used is indicated on the left, and the sizes of the protein markers are indicated on the right. Data in (**a, c**) represent at least 3 biological experiments. **d** Fold change (log₂) in BRM occupancy

between WT and *bcl7a bcl7b* background. Differentially occupied sites with a false discovery rate (FDR) < 0.05 are highlighted in purple (decreased sites) and green (increased sites). Two biological replicates were performed. Metagene plot (**e**), heatmap (**f**) and box plots (**g**) representation of the mean density BRM occupancy at all BRM binding sites in the WT and *bcl7a bcl7b* background. $n = 7465$. In box plots, center line and bounds of box represent median value and the interquartile range (IQR), respectively. Whiskers extend within 1.5 times the IQR. *P* values were calculated with the two-tailed Student's *t* test. **h** IGV screenshots showing BRM occupancy on selected loci in the WT and *bcl7a bcl7b* background. **i** Validation of ChIP-seq signals by ChIP-qPCR. *TA3* serves as a negative control. Error bars are presented as mean ± s.d. from three biological replicates. *P* values were calculated with the two-tailed Student's *t* test. Source data are provided as a Source Data file.

We then queried whether BCL7A/B are necessary for BRM targeting to chromatin genome-wide. ChIP-seq analyses showed that the binding peaks and corresponding genes of BRM were nearly identical in the WT and *bcl7a bcl7b* double mutant (Supplementary Fig. 7c). Indeed, among the 12,523 BRM-binding sites, only 787 sites showed significant changes in the absence of BCL7A/B, corresponding to 447 increases and 340 decreases (Fig. 3d). Furthermore, the average occupancy intensity of BRM-GFP was also almost the same in the *bcl7a bcl7b* mutant background compared with in the WT (Fig. 3e–g). These results were exemplified at individual loci (Fig. 3h and Supplementary Fig. 7d, e) and validated by ChIP-qPCR assay (Fig. 3i and Supplementary Fig. 7f).

To further validate the above results, we performed a spike-in ChIP-seq assay using the *Drosophila* chromatin. After spike-in

calculation, the average BRM-GFP binding signal at the BRM target peaks was similar between WT and the *bcl7a bcl7b* mutants (Supplementary Fig. 7g, h). Taken together, these data collectively demonstrate that BCL7A/B do not play a major role in the assembly and genomic localization of the BAS complexes.

## BCL7A/B loss results in impairment of BRM-driven chromatin accessibility

In yeast and mammals, previous studies have shown that BRG1—an ATPase homologous to plant BRM—is required for the rapid establishment of chromatin accessibility at SWI/SNF-complex target sites[2,3,38]. We therefore sought to assess the potential role of BCL7A/B subunits in mediating the remodeling activity of the complexes through an assay for transposase-accessible chromatin with

sequencing (ATAC-seq) comparing the *brm* and *bcl7a bcl7b* mutants with WT plants. The correlation between our ATAC datasets and other seedling ATAC datasets ($r = 0.92$) was very high (Supplementary Fig. 8a–d), and the FRIP values for WT samples were over 0.38 (Supplementary Data 2), exceeding the recommended threshold for ATAC-seq analysis[39], demonstrating the reproducibility of our assay. We identified 11,644 accessible regions in 14-day-old WT seedlings. The mean accessibility level at these regions was substantially lower in the *brm-1* null mutant than in the WT (Fig. 4a). Moreover, *brm-1* mutants had substantially reduced accessibility relative to the WT at 4762 regions (41% of accessible sites), whereas only 489 regions (4.2% of accessible sites) showed increased accessibility, indicating that the

majority of significantly changed sites displayed decreases in accessibility upon loss of BRM (Fig. 4b–d). BRM-occupied sites overlapped highly with accessible regions in the WT (Supplementary Fig. 8e), and both were enriched near the TSS (Supplementary Fig. 8f). Importantly, accessible regions occupied by BRM showed substantially reduced accessibility in the *brm-1* null mutant compared to the WT, suggesting that BRM is directly involved in promoting nucleosome accessibility in plants (Supplementary Fig. 8g). Together, these results show BRM complexes as major contributors shaping the genome-wide chromatin accessibility landscape in plants[16].

We then examined whether BCL7A and BCL7B are required for BRM-mediated chromatin accessibility alterations. The *bcl7a bcl7b*

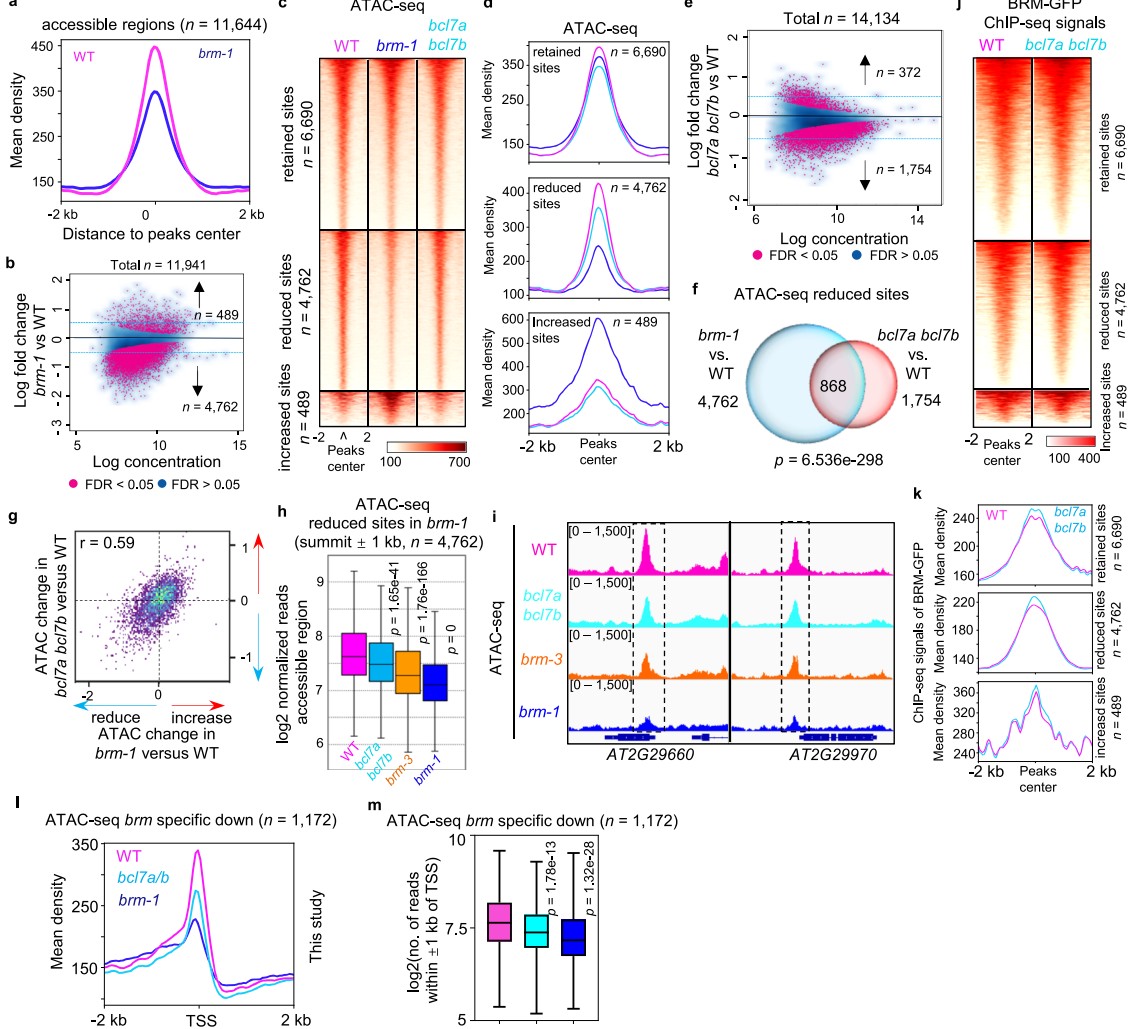

**Fig. 4 | Loss of BCL7A/B impairs BRM-driven chromatin accessibility.**
**a** Metagene plot reflecting substantially diminished accessibility at 11,644 accessible sites in WT upon loss of BRM. **b** Scatter plot showing fold-change of accessible peaks between WT and *brm-1*. Blue dots, stable peaks; pink dots, differential peaks. The numbers of differentially accessible peaks (increased or decreased) according to FDR are indicated. Heatmap (**c**) and metagene plots (**d**) reflecting the ATAC-seq signals over the retained, reduced, or increased chromatin accessibility sites of *brm-1* in WT, *bcl7a bcl7b*, and *brm-1*. **e** Scatter plot showing fold-change of accessible peaks between WT and *bcl7a bcl7b*. Blue dots, stable peaks; pink dots, differential peaks. The numbers of differentially accessible peaks (increased or decreased) are indicated. **f** Venn diagram showing overlap between the sites with reduced accessibility in *brm-1* compared with WT and those in *bcl7a bcl7b*. P values were calculated by the hypergeometric tests. **g** Scatter plot showing the correlation of chromatin accessibility changes (log$_2$(fold change)) between *brm-1* (x-axis) and *bcl7a bcl7b* (y-axis) at a merged set of all accessible regions. R values are indicated.

In both mutants, most affected regions show a comparable decrease in accessibility. **h** Box plots showing read counts at regions that showed reduced accessibility in *brm-1* for the indicated ATAC-seq experiments. In box plots, center line and bounds of box represent median value and the interquartile range (IQR), respectively. Whiskers extend within 1.5 times the IQR. P values were calculated with the two-tailed Student's t test. **i** Examples of ATAC-seq tracks at representative loci in the WT, *bcl7a bcl7b*, *brm-3* and *brm-1* mutants. Heat map (**j**) and metagene plots (**k**) reflecting BRM occupancy signals over sites that retained, reduced, or increased chromatin accessibility in *brm-1* upon depletion of BCL7A/B. Profile plots (**l**) and Box plots (**m**) showing the mean accessibility signals of the indicated samples at the genes with decreased accessibility specifically identified in *brm-1*. In box plots, center line and bounds of box represent median value and the interquartile range (IQR), respectively. Whiskers extend within 1.5 times the IQR. P values were calculated with the two-tailed Student's t test. Source data are provided as a Source Data file.

double mutant exhibited a significant increase in accessibility at fewer regions ($n = 372$), whereas many more regions ($n = 1754$) substantially decreased accessibility (Fig. 4e). This suggests that BCL7A/B mainly contributed to chromatin openness, similar to BRM. Regions that showed reduced accessibility in the *bcl7a bcl7b* mutant and those in the *brm-1* mutant showed significant overlap (Fig. 4f). However, roughly half of the downregulated accessible sites in the *bcl7a bcl7b* mutants are not observed in the *brm-1* mutants (Fig. 4f), suggesting that BCL7A/B have roles outside of the BAS complexes. The accessibility changes in the *bcl7a bcl7b* and *brm-1* mutants were positively correlated ($r = 0.59$; Fig. 4g). Genes with decreased DNA accessibility in the *bcl7a bcl7b* double mutant were enriched in Gene Ontology (GO) pathways response to light, response to hormone-mediated signaling pathway, tissue development, and leaf development, many of which were top pathways impacted by BRM mutation (Supplementary Fig. 9). Moreover, when we examined accessibility over regions that showed reduced accessibility in the *brm-1* null mutants, we found that loss of BCL7A/B also resulted in a significant reduction of accessibility at those regions compared with the WT (Fig. 4c, d, h), which were exemplified at individual loci (Fig. 4i). The degree of the reduction of DNA accessibility in the *bcl7a bcl7b* mutant was comparable to that in the *brm-3* hypomorphic allele (Fig. 4h, i). Finally, in comparing the binding levels of BRM-complexes in the WT and *bcl7a bcl7b* mutant over sites with reduced DNA accessibility, we found that the average binding intensity of BRM-GFP was nearly identical between the *bcl7a bcl7b* mutants and WT (Fig. 4j, k). Altogether, these data support a role for BCL7A/B in mediating BAS-complex-driven chromatin accessibility rather than complex stability and genome-wide complex localization.

Interestingly, when we compared the ATAC-seq of the *bcl7a bcl7b* mutants with those of *brip1 brip2* mutants, we found that there was significant overlapping between them as well as specifically mis-regulated accessible genes in the *bcl7a bcl7b* mutants that are not found in *brip1 brip2* (Supplementary Fig. 10a), consistent with the notion that BCL7A/B play roles outside of the BAS complexes. In support of this, we observed that genes exhibit changed chromatin accessibility in the *syd* or *minu* mutants[16] and those in *bcl7a bcl7b* double mutants (this study) showed significant overlap (Supplementary Fig. 10b).

To further evaluate the role of BCL7A/B in BAS-driven accessibility regulation, we examined genes that display a reduction of accessibility specific in *brm-1* mutants (no reduction in the *syd* or *minu* mutants) (Supplementary Fig. 10c)[16] and found that loss of BCL7A/B still resulted in a significant decrease of accessibility (Fig. 4l, m). This reduction upon the loss of BCL7A/B was comparable to the decrease in the *brip1 brip2* double mutants (Supplementary Fig. 10d, e). Together, these data indicate that BCL7A/B are specifically responsible for BAS-driven chromatin accessibility at these genes. Likewise, loss of BCL7A/B caused a significant downregulation of accessibility at genes with a specific decrease in accessibility in *syd* or *minu* (Supplementary Fig. 10f–k), suggesting that BCL7A/B are likely involved in regulating the accessibility remodeling ability of the SAS and MAS complexes.

Finally, we carried out MNase-seq assays on the WT, *brm-1*, and *bcl7a bcl7b* mutants. We found that the abundance of the nucleosome at genes that specifically require BRM for chromatin accessibility was increased in *brm-1* and, to a lesser extent in *bcl7a/b* (Supplementary Fig. 11). These data indicate that nucleosomes are still present in the absence of BCL7A/B, but the BRM-mediated nucleosome accessibility at these genes is impaired. It is known that nucleosomes have inhibitory effects on chromatin accessibility and gene expression. Therefore, the MNase-seq data also suggest that the BAS complexes may remove or loosen nucleosomes from chromatin to maintain chromatin accessibility.

## BCL7A/B and BRM co-regulate gene expression

We next investigated the contribution of BCL7A/B to the transcriptional regulation of BAS-complex target genes by performing an RNA-sequencing (RNA-seq) analysis using a set of single, double, and triple mutant seedlings (Supplementary Fig. 12a and Supplementary Data 3). Compared with the WT, the *bcl7a bcl7b* mutant had 3358 upregulated and 2877 downregulated genes, a large percentage of which shared the same direction of mis-regulation in the *brm-1* single mutant (Fig. 5a). Hierarchical clustering and correlation analyses across the WT and different mutants revealed that the transcriptome profile of the *bcl7a bcl7b* mutant showed similarity to those of the *brm-3* and *brm-1* mutants (Fig. 5b, c). Consistent with this, the transcriptome of the *bcl7a bcl7b* mutant was positively correlated with that of the *brm-1* null mutant (correlation coefficient $r = 0.78$; Fig. 5d). In contrast to genes unaffected in the *bcl7a bcl7b* mutant, both *bcl7a bcl7b*-downregulated and -upregulated genes were also down- and up-regulated, respectively, in *brm-1*, *brm-3*, *brm-3 bcl7a bcl7b*, and *brm-1 bcl7a bcl7b* mutant plants (Fig. 5e and Supplementary Fig. 12b). Similarly, *brm-1*-downregulated and -upregulated genes but not *brm-1*-unaffected genes showed downregulation and upregulation, respectively, in *bcl7a bcl7b* double mutants (Supplementary Fig. 12c, d). Furthermore, relative to *bcl7a bcl7b*-unaffected or -upregulated genes, *bcl7a bcl7b*-downregulated genes showed significant enrichment in ChIP-seq signals for BCL7A/B, BRM, and H4K5ac, but not H3K27me3 (Fig. 5f, g and Supplementary Fig. 12e). Additionally, as was the case with *bcl7a bcl7b*-downregulated genes, *brm-1*-downregulated genes were enriched for BCL7A/B, BRM, and H4K5ac ChIP-seq signals, but not H3K27me3 (Supplementary Fig. 12f, g). We also compared the RNA transcriptome of *bcl7a bcl7b* with those of *brip1 brip2* and *brd1 brd2 brd13*. There was significant overlapping as well as specifically mis-regulated genes in *bcl7a bcl7b* that were not found in *brip1 brip2* or *brd1 brd2 brd13* (Supplementary Fig. 12h, i), suggesting that BCL7A/B also play transcriptional regulation functions independent of the BAS complexes.

Finally, integration of ATAC-seq and RNA-sequencing (RNA-seq) datasets identified a subcluster of genes that showed decreased gene expression and for which nearby DNA accessibility, but not BRM occupancy, was strongly reduced in the *bcl7a bcl7b* mutant relative to the WT (Fig. 5h, i). GO term analysis of these sites revealed terms related to various developmental and response processes, such as response to light, response to lipid, response to fungus, and shoot system development, among others (Fig. 5j). Together with the observations that the *bcl7a bcl7b* double mutation did not affect the enrichment of BRM at target genes but reduced the BRM-driven chromatin accessibility (Figs. 3 and 4), our data suggest that BCL7A/B regulate gene transcription through potentiating DNA accessibility regulation ability of the SWI/SNF complexes.

## BCL7A/B are required for juvenile identity by facilitating BRM-mediated chromatin accessibility at *MIR156A/C*

Having established the mechanism through which BCL7A/B regulate genome-wide gene expression in the context of the BRM complex, we finally sought to determine the biological significance of this mechanism by examining the phenotypic consequences of *BCL7A/B* mutation. The *bcl7a bcl7b* double mutants did not show typical morphological phenotypes such as short root caused by *BRM* mutation[16], but we noticed that the leaves of *bcl7a bcl7b* double mutants appeared larger and more elongated under long-day (LD) conditions (Supplementary Fig. 13). In Arabidopsis, juvenile leaves are small, round and, lack trichomes on their abaxial surface, whereas adult leaves are larger and elongated, with serrated margins, and produce abaxial trichomes[40]. Thus, *bcl7a bcl7b* seedlings seemed to have a precocious adult phenotype compared with the WT, suggesting that BCL7A/B might be negative regulators of the juvenile-to-adult phase transition. To explore this, we examined *bcl7a*, *bcl7b*, and *bcl7a bcl7b* mutant seedlings under short-day (SD) conditions. The initiation of abaxial trichomes was accelerated in *bcl7b* single mutants, whereas *bcl7a* single mutants had no discernible difference from the WT in this regard (Fig. 6a). Loss of BCL7A enhanced the phenotypes of *bcl7b*

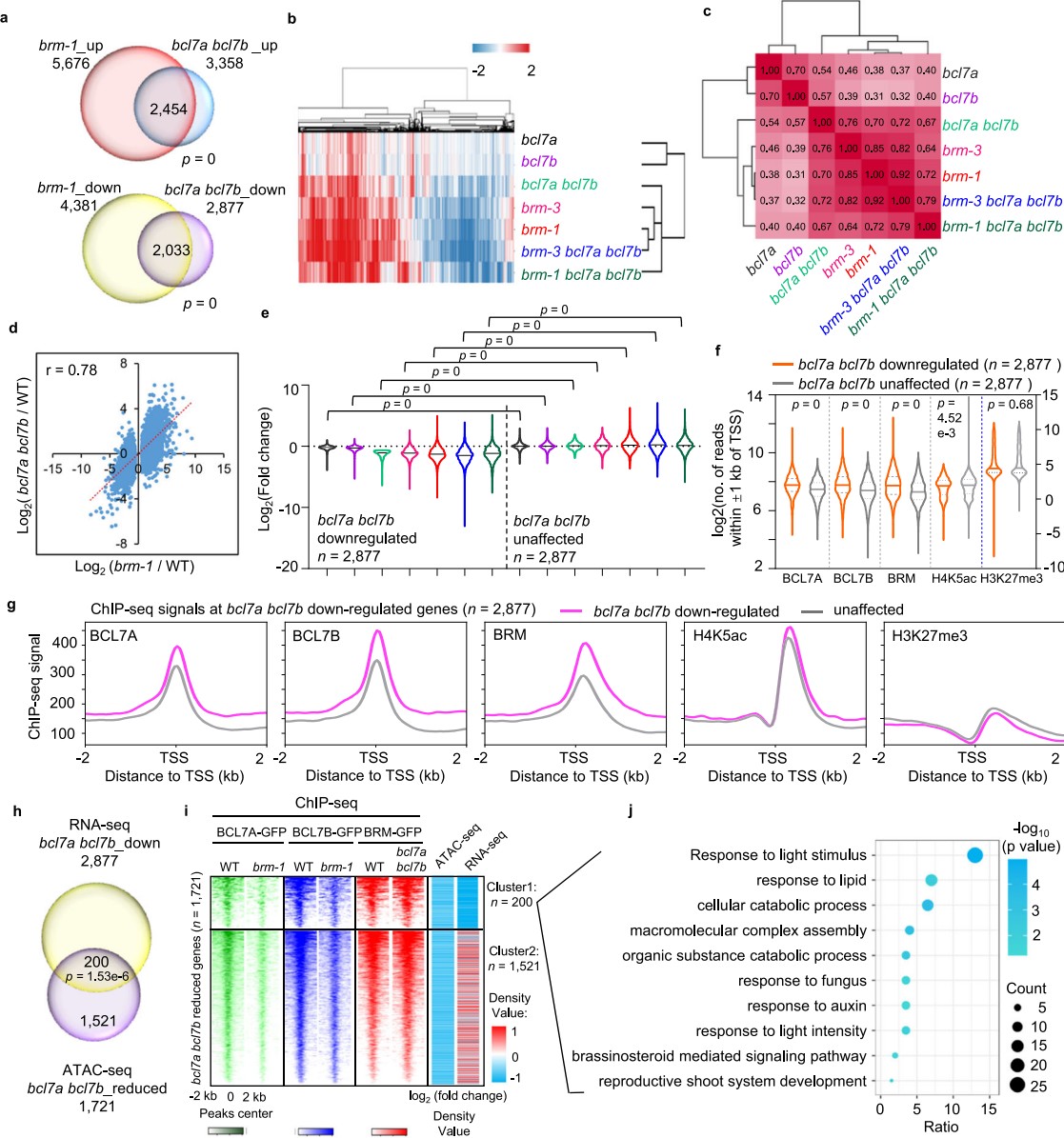

**Fig. 5 | BCL7A/B and BRM co-regulate gene expression. a** Venn diagrams showing statistically significant overlaps between genes up- or down-regulated in *brm-1* and those in *bcl7a bcl7b*, according to the hypergeometric test. **b** Heatmap showing hierarchical clustering of differentially expressed genes in different mutants. Red and blue represent up- and down-regulation in mutants, respectively (total mis-regulated genes, *n* = 15,924). **c** Matrix depicting Pearson correlation coefficients among RNA-seq datasets. Adjusted *r* values were indicated. **d** Scatter plot showing positive correlation at genes that are differentially expressed in *brm-1* between *bcl7a bcl7b* and *brm-1*. The red line indicates the line of best fit, and the *r* value was indicated. **e** Violin plots depicting the log₂(fold change) in RNA-seq for *bcl7a bcl7b* downregulated and unaffected genes in various mutants. *p* values are from two-tailed Mann–Whitney *U* test. **f** Violin plots displaying read counts at *bcl7a bcl7b* downregulated and unaffected genes for the indicated ChIP-seq data. Reads were summed ±1 kb from the TSS. *p* values are from two-tailed Mann-Whitney *U* test. **g** Average binding signals of the indicated ChIP-seq at *bcl7a bcl7b*-downregulated or *bcl7a bcl7b*-unaffected genes. Read counts per gene were summed in 50-nt bins. **h** Venn diagrams showing significant overlaps between the genes downregulated in *bcl7a bcl7b* and genes loss chromatin accessibility in *bcl7a bcl7b*. *P* values were calculated by the hypergeometric tests. **i** Heat map displaying BCL7A, BCL7B and BRM occupancy levels at sites loss chromatin accessibility in *bcl7a bcl7b*. log₂(fold change) values of accessibility and nearest gene expression in *bcl7a bcl7b* relative to WT were also displayed. Data were divided into two clusters according to (**h**). **j** Gene ontology analysis using 200 genes nearest to Cluster 1 sites in (**i**). *P* values were calculated by the Fisher's exact test. Source data are provided as a Source Data file.

mutants, with the *bcl7a bcl7b* double mutant exhibiting earlier abaxial trichome production than the *bcl7b* mutant (Fig. 6a). These results recapitulate the results of a recent study examining the accelerated vegetative phase transition in the absence of BRM[41]. Consistent with this, *bcl7a bcl7b* double mutants produced leaves that were significantly more adult in shape (i.e., longer and narrower) than WT leaves: the leaf blade length: width ratio of the 7th leaf of the *bcl7a bcl7b* mutant was significantly higher than that of the 7th leaf of the WT, whereas the leaf blade length: width ratio of the 7th leaf of the

*bcl7a* mutant was not significantly different from that of the 7th leaf of the WT (Supplementary Fig. 14a). The precocious transition phenotype of *bcl7a bcl7b* double mutant seedlings was rescued by a *BCL7A-GFP* or *BCL7B-GFP* transgene, confirming that the precocious adult phenotype was caused by the lack of BCL7A/B and that the transgenic materials are functional in vivo (Fig. 6a, b and Supplementary Fig. 14a). Together, these results indicate that BCL7A and BCL7B have redundant functions, with the later having a more important role in maintaining juvenile identity.

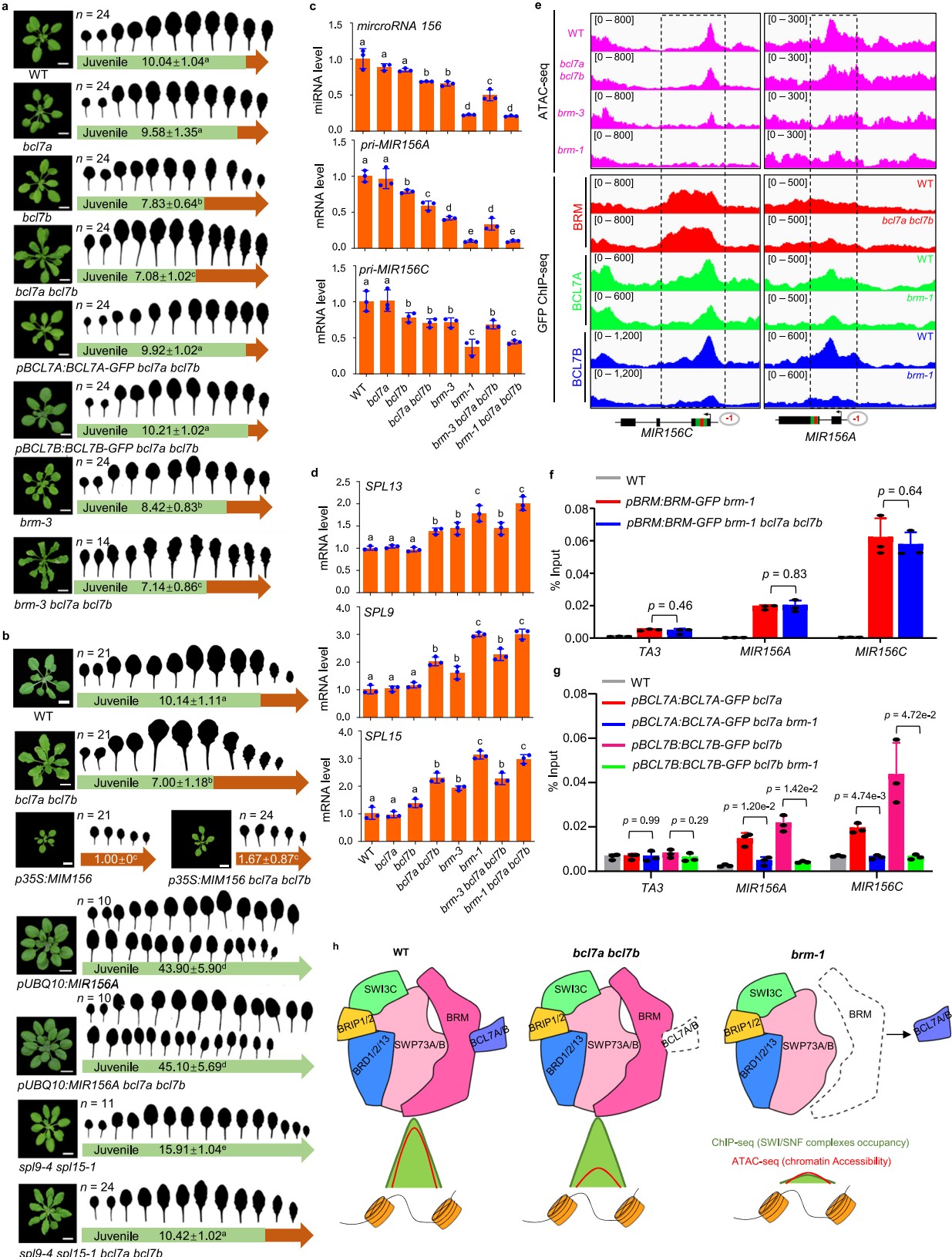

Plant juvenility is controlled by high levels of miR156[24,27–29]. To determine whether the accelerated vegetative phase change of the *bcl7a bcl7b* mutant is associated with changes in the level of miR156, we performed RT-qPCR and showed that the accumulation of mature miR156 was significantly reduced in the *bcl7a bcl7b* mutant (Fig. 6c). The abundance of the primary transcripts of *MIR156A* and

*MIR156C* (*pri-MIR156A/C*), both of which are highly expressed *MIR156* loci contributing to the level of mature miR156[42,43], were also decreased significantly in the *bcl7a bcl7b* and *brm-1* mutants (Fig. 6c and Supplementary Fig. 14b, c). A decrease of *pri-MIR156A/C* was also observed in *bcl7b* single mutants, but to a lesser extent than in the double mutants, consistent with the functional redundancy between

**Fig. 6 | BCL7A/B negatively regulate vegetative phase transition through facilitating BRM-mediated chromatin accessibility at *MIR156A/C*. a, b** Vegetative phase transition phenotypes of WT and various mutants under SD condition. Leaf shape of fully expanded rosette leaves of different genotypes in 21-day-old was shown. The color in the arrow indicates juvenile leaves (green) and adult leaves (brown). The number ± SD indicates the appearance of the first leaf with abaxial trichomes. Lowercase letters indicate significant differences between genetic backgrounds, as determined by the *post hoc* Tukey's HSD test. *n* indicates the number of plants used. Scale bars, 1 cm. **c, d** Relative expression of *pri-MIR156A*, *pri-MIR156C*, *SPL13*, *SPL9*, and *SPL15* in 14-day-old seedlings grown in SD conditions. *ACTIN2* served as the internal control. The relative accumulation of mature miR156 was determined by stem-loop RT-qPCR. The small nucleolar RNA *snoR101* served as the internal control. Mean ± s.d. from three biological replicates. Lowercase letters indicate statistical significance determined by the *post hoc* Tukey's HSD test. **e** IGV views of ATAC-seq and ChIP-seq signals over *MIR156A/C* in the indicated plant materials. **f, g** Validation of the occupancy at *MIR156A/C* by ChIP-qPCR in the indicated transgenic plants. *TA3* serves as a negative control. Mean ± s.d. from three biological replicates. Statistical significance was determined by two-tailed Student's *t* test; **h** Model of genome-wide BRM-containing SWI/SNF complex occupancy (ChIP-seq) and chromatin accessibility (ATAC-seq) in WT, *bcl7a bcl7b*, and *brm-1* at BCL7A/B-driven BRM-SWI/SNF complex sites. Source data are provided as a Source Data file.

BCL7A and BCL7B (Fig. 6c). To further validate that BCL7A/B upregulate the level of miR156 at the transcriptional level, we crossed a GUS reporter line under the promoter of MIR156A (*pMIR156A:GUS*)[43] into the *bcl7a bcl7b* background. GUS staining and RT-qPCR showed a substantial reduction in the expression of *GUS* in the *bcl7a bcl7b* mutant compared with that in the WT (Supplementary Fig. 14d, e), suggesting that BCL7A/B activate miR156 accumulation by promoting the transcription of the *MIR156A* locus in seedlings. miR156 directly represses the expression of *SQUAMOSA PROMOTOR BINDING PROTEIN-LIKE* (*SPL*) genes, among which *SPL9*, *SP13*, and *SPL15* play dominant roles in the vegetative phase transition[44]. We found that the expression levels of *SPL9*, *SPL13*, and *SPL15* were unaffected in the *bcl7a* mutant but increased slightly in the *bcl7b* mutant and even more so in the *bcl7a bcl7b* mutant, corresponding to the levels of miR156 in these mutants (Fig. 6d). Together, these results reveal that BCL7A/B have an important role in activating *MIR156A/C* in juvenile seedlings.

We next assessed if BCL7A/B maintain juvenility by promoting the accumulation of miR156, which downregulates *SPLs*. We crossed *p35S:MIM156*, *pUBQ10:miR156A*, and *spl9-4 spl15-1* with *bcl7a bcl7b* and examined their effect on the phenotype of the *bcl7a bcl7b* mutant (Fig. 6b and Supplementary Fig. 14f). The *p35S:MIM156* plants constitutively expressed a transcript with a non-cleavable miR156 target site (a target-site mimic, MIM156) that acts through sequestration of miR156 and therefore prematurely produced adult features since the first leaf[45]. We found that, in contrast to the WT background, *bcl7a bcl7b* double mutations in *p35S:MIM156* had no additional effect on the accelerated onset of the adult phase, showing that BCL7A/B act through a MIR156-dependent pathway (Fig. 6b, c). Overexpression of *MIR156A* in *pUBQ10:miR156A* completely suppressed the accelerated abaxial trichome production of the *bcl7a bcl7b* mutant, suggesting that the effect of the *bcl7a bcl7b* mutation on the juvenile-to-adult transition is attributable to their repression on miR156 (Fig. 6b, c). Consistent with these interpretations, knocking out of *SPL9* and *SPL15* (*spl9-4 spl15-1*) partially inhibited the accelerated adult transition of the *bcl7a bcl7b* mutant, confirming that BCL7A/B maintain juvenility by repressing the expression of *SPL* genes through promoting *MIR156* transcription (Fig. 6b, c).

To examine the functional relationship between BCL7A/B and BRM in regulating the juvenile-to-adult transition, we examined the *brm-3 bcl7a bcl7b* and *brm-1 bcl7a bcl7b* mutants. *brm-3* is a hypomorphic allele, whereas *brm-1* is a null mutant. Under SD conditions, the *brm-3* mutant produced abaxial trichomes on leaf 8.4, which is significantly earlier than in the WT, whereas the *bcl7a bcl7b brm-3* triple mutant produced abaxial trichomes on leaf 7.1. Thus, the early production of abaxial trichomes of the *brm-3* hypomorphic mutant was further accelerated by the introduction of the *bcl7a bcl7b* mutations (Fig. 6a). The leaves of the *brm-1* null mutant were extremely downward curled, making examining its leaf shape and trichome initiation phenotypes infeasible to determine the effect of loss of BRM on vegetative phase change. Nevertheless, we found that the *bcl7a bcl7b* mutations did not exacerbate the downregulation of miR156 and upregulation of *SPL* genes in the *brm-1* mutant (Fig. 6c, d), indicating

that BCL7A/B and BRM operate in the same pathway to regulate the juvenile-to-adult transition.

It was recently reported that DNA accessibility levels in the JRR upstream of the *MIR156A/C* TSS are correlated with their transcriptional activity; the *MIR156A/C* JRRs in juvenile leaves are much more accessible than those in adult leaves[30]. However, the factors and mechanisms making the JRRs accessible in juvenile leaves are unknown. We found that while the JRRs of *MIR156A/C* were highly accessible in WT seedlings, their chromatin accessibility was nearly abolished in the null *brm-1* mutant seedlings (Fig. 6e and Supplementary Fig. 14g), demonstrating that the open chromatin state of *MIR156A/C* in the juvenile stage is essentially dependent on the BRM-containing SWI/SNF complexes. Moreover, compared with the WT seedlings, the *bcl7a bcl7b* and *brm-3* mutants displayed reduced accessibility at the JRRs (Fig. 6e and Supplementary Fig. 14h), consistent with the reduced chromatin remodeling activity of the BRM-complex in these two mutants. Finally, the bindings of BRM at *MIR156A/C* were not affected by mutation of BCL7A/B, whereas the bindings of BCL7A/B at *MIR156A/C* depended on BRM (Fig. 6f, g), consistent with the ChIP-seq data (Fig. 6e). Notably, the accessibility at *MIR156A* and *MIR156C* was not reduced in *syd* and *minu* mutants (Supplementary Fig. 15)[16], suggesting that these two loci are specifically regulated by BCL7A/B in the BAS complexes. Taken together, these data demonstrate that BAS-complex-mediated chromatin remodeling, facilitated by BCL7A and BCL7B, is crucial for the transcriptional activation of *MIR156A/C* during juvenile development through opening the local chromatin.

## Discussion

Mutations in genes encoding subunits of SWI/SNF chromatin remodeling complexes often cause severe developmental disorders and misregulation of global gene expression in eukaryotes including plants[8,46–51]. However, the subunit-specific contributions to SWI/SNF complexes function are far from clear, presenting major barriers to our mechanistic understanding of the biological effects of mutations on SWI/SNF complexes. Although chromatin accessibility is critically important in vivo, little is known regarding subunits that regulate this function within SWI/SNF remodeler complexes. Of note, although BCL7s have been shown as subunits of SWI/SNF complexes for decades in animals and recently in plants, the role of BCL7 subunits in SWI/SNF function remains elusive. In this study, we have defined a unique functional role for Arabidopsis BCL7A and BCL7B in modulating BAS-complex-mediated genome-wide chromatin accessibility without changing the complex assembly and genomic localization (Fig. 6h). We further show that the BAS complexes have a critical role in generating DNA accessibility at *MIR156A/C* promoters and that this ATPase-driven DNA accessibility is potentiated by BCL7A/B for regulating the vegetative phase transition in plants.

Loss of BCL7A/B only compromises but does not completely destroy BAS remodeling activity. Thus, we propose that BCL7 subunits are likely evolved for fine-tuning the SWI/SNF activity to a suitable level at target genes for optimized gene transcription regulation in multicellular eukaryotes. Unexpectedly, we found that the BAS complexes

lacking BCL7A/B retain the ability to assemble correctly and target chromatin genome-wide, but they are defective in generating chromatin accessibility and regulating critical target genes, such as *MIR156A/C*. These findings represent the first time in eukaryotes in which we have identified subunits that only regulate the SWI/SNF remodelers' activity but do not affect their assembly and genomic localization. Interestingly, these data present a contrast to our recent studies examining the role of BRIP1/2 or BRD1/2/13 subunits, in that complexes lacking BRIP1/2 or BRD1/2/13 are both unable to assemble, and target to genes[14,15]. Together, our complementary observations underscore the importance of comprehensively investigating the function of each subunit to disconnect the role of chromatin accessibility regulation from SWI/SNF complex assembly and genomic targeting. It is worth noting that although our combined omics experiments (ATAC-seq and ChIP-seq) have provided compelling evidence supporting the function of BCL7A/B in fine-tuning BAS activity, in vitro remodeling activity experiments using intact BAS complexes could provide further insights into the function of BCL7s. Furthermore, whether BCL7s also affect the activity of SAS and MAS complexes without influencing their assembly requires further validation.

In our study, we observed a relatively mild overlap between the misregulated genes identified by RNA-seq and ATAC-seq in *bcl7ab* mutants. The overlap of down-regulated genes between ATAC-seq and RNA-seq in the *brm-1* mutant was comparable to that observed in *bcl7a bcl7b* mutants. Two reasons might explain the significant yet mild overlapping observed in the *bcl7* mutants. First, it is argued that chromatin accessibility change is necessary, but not sufficient, for gene transcription based on evidence from studies in various systems. Current evidence in the field argues for a multilayered regulatory program in which chromatin accessibility plays a broad licensing role but does not determine the gene expression state[1,52]. Second, the loss of BCL7A/B compromises but does not completely destroy the BAS-mediated chromatin accessibility changes. Therefore, the mild changes in accessibility caused by the loss of BCL7 might account for the mild changes in gene expression. BRM, in contrast, serves as the core ATPase within the BAS complexes; its absence should completely destroy the remodeling activity, leading to a more pronounced impairment of chromatin accessibility and gene transcription compared to the loss of BCL7.

Notably, the BCL7A/B orthologs in humans were suggested to be subunits of human SWI/SNF complexes and frequently undergo biallelic inactivation in diffuse large B-cell lymphomas[34,35]; however, structural information about human BCL7A/B/C in the assembled cBAF complex is unclear, and their specific roles and therefore the effects of their mutations on human SWI/SNF complex function are completely unknown[53,54]. Interestingly, loss of BCL7A/B does not entirely disrupt but only downregulates the BAS-mediated chromatin accessibility regulation. Thus, our work pinpointing the role of these evolutionarily conserved subunits frequently perturbed in human cancer may suggest exciting therapeutic opportunities for chemically fine-tuning SWI/SNF remodeling activities through targeting BCL7A/B/C proteins.

The levels of miR156 are dynamically regulated during the plant life cycle and correlated with plant juvenile and adult stages: miR156 is highly expressed in juvenile seedlings, decreases in adult seedlings, and is re-activated in embryos. Recently, the seed-specific transcription factor LEC2 was shown to be required for the de novo re-activation of *MIR156A/C* in embryos through its promotion of chromatin accessibility at the RY (CATGCA) cis-element of the JRR region in the promoters[30]. Surprisingly, loss of *LEC2* not only compromises *MIR156A/C* de novo expression in embryos but also leads to *MIR156A/C* downregulation in the juvenile phase accompanied by a shortened juvenile phase. These observations indicate that the open chromatin state triggered by LEC2 is transmitted from an embryo to post-embryonic stages even in the absence of *LEC2* expression later in seedlings, suggesting a memory of embryonic *MIR156A/C* activation

status in the juvenile phase. Since LEC2 is an embryonic-specific factor that is not expressed in other tissues under normal conditions, a critical question is how this embryonic open chromatin state is memorized in juvenile seedlings. Our results demonstrated that BCL7A/B and BRM, as the core subunits of BAS complexes, function to configure a highly accessible chromatin environment at the JRRs of *MIR156A/C* in the juvenile seedlings that sustains a high level of *MIR156A/C* expression, revealing the molecular mechanism through which juvenile identity is ensured. In addition, as a DNA-binding transcription factor, LEC2 could not mediate chromatin accessibility by itself alone but must depend on other factors. Because BCL7A/B and BRM are ubiquitously expressed throughout the whole life cycle of Arabidopsis, including in embryos, it will be interesting to test whether BCLA/B-BRM-mediated chromatin remodeling activity may be employed, directly or indirectly, by LEC2 to create an active accessible chromatin environment at *MIR156A/C* in embryos for the juvenility resetting in each generation. On the other hand, *MIR156A/C* expression declines through adult phases. How does the BAS complex maintain *MIR156A/C* expression in the juvenile stage but not in the adult stage? Previous studies showed that POLYCOMB REPRESSIVE COMPLEX2 (PRC2) represses *MIR156A/C* through depositing repressive histone mark H3K27me3[30,50,55]. The decrease of *miR156* expression with age is temporally correlated with a gradual increase in the level of H3K27me3 and the accumulation of PRC2 complex at the *MIR156A/C* loci following plant growth. It is known that BRM and PRC2 act antagonistically at the nucleosome level to regulate gene expression in plants[47,56]. Therefore, the progressive increase of PRC2 complex and H3K27me3 at the *MIR156A/C* loci with age may result in a closed nucleosome status that antagonizes the activity of BAS complex at the *MIR156A/C* in the adult stage.

*SPL9* and *SPL15* were recently shown to be the two major players in regulating vegetative phase transition[44]. We observed that BRM and BCL7A/B bind to *SPL9* and *15* (Supplementary Fig. 16). ATAC-seq data showed a decrease in chromatin accessibility of *SPL9/15* in both *brm* and *bcl7a/b* mutants compared to wild-type (Supplementary Fig. 16). However, albeit decreased chromatin accessibility at *SPL9/15*, the mRNA levels of *SPL9/15* were actually increased in *brm* and *bcl7a/b* mutants (Fig. 6d). This increase in mRNA levels of *SPL9/15* is because of the reduced *miR156* levels. Therefore, these data do not support the notion that BAS complex directly regulates the *SPLs* bypassing *miR156*.

In addition to being associated with juvenility resetting in embryos, widespread reprogramming of chromatin accessibility has also been functionally linked to other developmental transitions and to responses to diverse internal or external stimuli in plants, including pollen formation during the diploid-to-haploid transition, leaf cell differentiation, root cell identity, responses to phosphate starvation, and responses to cytokinin or auxin[52,57–66]. For example, thousands of accessible chromatin regions specific to pollen were found during the diploid-to-haploid transition and strongly associated with functions unique to pollen development[58]. Many quantitative differential chromatin accessibility regions between shoot stem cells and differentiated leaf mesophyll cells or between vasculature and epidermis cells in LD conditions were also identified[60,62]. Likewise, differential chromatin accessibility regions among different root cell types were found[63,64]. In addition, a genome-wide alteration of chromatin accessibility in response to phosphate starvation was observed in root cells[57]. Auxin was found to promote the acquisition of plant cell totipotency through reprogramming chromatin accessibility during somatic embryogenesis and shoot regeneration[52,66]. Finally, chromatin accessibility also undergoes extensive reprogramming in response to cytokinin to direct transcriptional regulation[59,66]. Overall, these studies highlighted the necessity of chromatin accessibility dynamics in establishing cell-type-specific or stimulus-responsive expression of relevant genes; however, the factors directly involved in the modulation of chromatin accessibility, and therefore the underlying mechanisms, are largely unknown.

Here, our ATAC-seq data showed that loss of BRM results in defects in generating WT levels of accessibility at thousands of genes, accounting for a large portion of the accessible regions in the WT[16]. Moreover, GO term analysis of the regions whose accessibility depends on BRM revealed many genes enriched in the processes of shoot development, shoot development, and responses to different stimuli such as light, wounding, and phytohormone (Supplementary Fig. 9b). Therefore, the BAS complex is a general regulator for chromatin accessibility reprogramming during diverse developmental transitions and in response to both internal and external signals. Together, our work lays the groundwork for further understanding how genome-wide chromatin accessibility is shaped during plant developmental phase transitions and in response to ever-changing environments.

Apart from BRM, three other ATPases, SYD, MINU1, and MINU2, were also co-immunoprecipitated with BCL7A-GFP and BCL7B-GFP, indicating that BCL7A/B are present in the SAS- and MAS-complexes (Fig. 2a–c and Supplementary Fig. 5a–c)[16,17,33]. Interestingly, the *bcl7a bcl7b* double mutant displayed less severe phenotypic and chromatin accessibility defects than the *brm*, *syd*, and *minu1 minu2* mutant. One interpretation for this is that the absence of BCL7A and BCL7B could affect the functions of all subtypes of plant SWI/SNF complexes without completely abrogating their activities, while the loss of the ATPase subunit could fully destroy the function of the complexes. BCL7A/B also play a role in fine-tuning the activity of SAS and MAS chromatin remodeler complexes (Supplementary Fig. 10). It will be necessary to examine whether BCL7A/B are not required for the assembly of the SAS and MAS complexes and/or association of them to chromatin.

In summary, our work identifies a specialized role for the conversed BCL7A and BCL7B subunits in regulating the chromatin remodeling activity of SWI/SNF complexes, thus decoupling the assembly and genomic binding of the complexes from their nucleosome remodeling activity and suggesting that targeting BCL7A/B/C proteins might be potential therapeutic approaches to chemically fine-tuning SWI/SNF remodeling activities for SWI/SNF-related tumors. Our results also highlight the role of SWI/SNF-mediated chromatin accessibility facilitated by BCL7A/B in juvenile memory and provide key insights into the mechanisms by which SWI/SNF subunit mutations cause developmental disorders.

## Methods
### Plant materials and growth conditions
Transfer DNA insertion lines, *bcl7a-1* (SALK_027934) and *bcl7b-1* (SALK_029285) were obtained from the Arabidopsis Biological Resource Center (ABRC) and were described recently[16]. Mutants *brm-1* (SALK_030046) and *brm-3* (SALK_088462) and *pBRM:BRM-GFP brm-1* transgenic plants were previously described[31]. The *spl9-4 spl15-1* (SAIL_150_B05 and SALK_074426), *p35S:MIM156*, and *pMIR156A:GUS* seeds were provided by Pro. Gang Wu (Zhejiang Agriculture and Forestry University) and previously described[29,41]. Primers used for genotyping are listed in Supplementary Data 4.

For RT-qPCR/RNA-seq, ChIP-qPCR/ChIP-seq, ATAC-seq, and IP-MS assays, Arabidopsis seeds were sterilized with 10% sodium hypochlorite solution, washed with sterile water four times, and then stratified at 4 °C in darkness for three days. Seeds were then sown on ½-strength Murashige and Skoog (MS) medium containing 1% sucrose and 0.6% agar. For phenotypic analysis, seeds were sown on a mixture of soil and vermiculite (1:1). Seedlings were grown under cold white light (approximately 120 μmol·m$^{-2}$·s$^{-1}$) with short-day conditions (10 h light/14 h dark) at 22 °C. Abaxial trichomes were observed with a stereomicroscope 3–4 weeks after planting. For leaf shape analysis, fully expanded leaves were removed, attached to cardboard with double-sided tape, flattened with transparent tape, and then scanned in a digital scanner. Rips in the leaf blade produced during this process were filled in using Photoshop.

### Generation of transgenic plants
To construct *pBCL7A:BCL7A-GFP* and *pBCL7B:BCL7B-GFP* vectors, the full-length genomic DNA of *BCL7A* or *BCL7B* (without the stop codon) containing its corresponding promoter sequence (~2.7 kb for *BCL7A* promoter and ~2.3 kb for *BCL7B* promoter) was amplified from genomic DNA by PCR, cloned into the *pDONR221* vector by BP reaction (Invitrogen,), and further subcloned into the destination plasmid *pMDC107*[67] by LR reaction (Invitrogen). The constructs were introduced into *Agrobacterium tumefaciens* strain *GV3101* and were then used to transform *bcl7a bcl7b* double mutant plants using the floral dip method[68]. The *pBCL7A:BCL7A-GFP bcl7a* was isolated after crossing *pBCL7A:BCL7A-GFP bcl7a bcl7b* with *brm-1*. The same strategy was used to obtain *pBCL7B:BCL7B-GFP bcl7b*.

To construct the vector overexpressing *MIR156A* precursors, a genomic DNA fragment containing *pre-MIR156A* was amplified from genomic DNA by PCR as previously described[24], and further subcloned into a modified overexpression vector *pFGC-UBQ10pro*[69] using ClonExpress II One Step Cloning Kit (Vazyme Biotech, Cat. No. C112). The construct was introduced into *Agrobacterium tumefaciens* strain *GV3101* and were then used to transform WT and *bcl7a bcl7b* double mutant plants using the floral dip method[68]. Primers used for constructing are listed in Supplementary Data 4.

### BiFC assays
BiFC vectors were constructed as previously described[14]. In brief, the full-length coding sequences of BCL7A and BCL7B were amplified from Col-0 cDNA and cloned into the *pEarleyGate 201-nYFP* or *pEarleyGate 202-cYFP* vector[70] by LR reaction (Invitrogen). The constructs were then individually introduced into *A. tumefaciens* strain *GV3101*, and the resulting bacteria were used to infiltrate the lower epidermal cells of *N. benthamiana* leaves[71]. After 48 h, YFP fluorescence signals were visualized using a confocal microscope (LSM880 with Fast Airy scan). HAT3 (encoded by *AT3G60390*) was used as a negative control. Sequences for the primers used are listed in Supplementary Data 4.

### Co-immunoprecipitation and mass spectrometry
For co-IP using tobacco transient expression system, full-length coding sequences of *BCL7A*, *BCL7B*, *SWI3C*, and *SWP73B* were amplified from Col-0 cDNA and cloned into the BamHI site of the *pHB-HA* or *pHB-FLAG* vector[15]. The vectors *pEAQ-BRM-N-GFP*, *pHB-BRIP2-HA*, and *pHB-BRD2-HA* were previously described[14,15]. All primers used for vector construction are listed in Supplementary Data 4. The constructs were then individually introduced into *A. tumefaciens* strain *GV3101*, and the resulting bacteria were used to infiltrate the lower epidermal of cells of *Nicotiana benthamiana* leaves[71]. Two days after infiltration, leaves were collected and homogenized in 1.5 ml of lysis buffer (50 mM HEPES [pH 7.5], 300 mM NaCl, 10 mM EDTA, 1% Triton X-100, 0.2% Nonidet P-40, 10% glycerol, 2 mM DTT, 1× Complete Protease Inhibitor Cocktail (Roche)) at 4 °C for 30 min. After centrifugation at 14,000 g and 4 °C for 15 min, the supernatant was diluted by equal volume dilution buffer (50 mM HEPES [pH 7.5], 10 mM EDTA, 10% glycerol, 2 mM DTT, 1× Complete Protease Inhibitor Cocktail (Roche)) and then incubated with anti-HA beads (Sigma, Cat. No. A2095) or anti-FLAG beads (Bimake, Cat. No. B26101-1ML) at 4 °C for 3 h with gently rotation. Beads were then washed three times with washing buffer (50 mM HEPES [pH 7.5], 150 mM NaCl, 10 mM EDTA, 0.2% Triton X-100, 0.1% Nonidet P-40, 10% glycerol). Proteins were eluted in SDS loading buffer and incubated at 55 °C for 10 min, followed by immunoblotting.

For co-IP of stable Arabidopsis transgenic plants, 6 g of 14-day-old seedlings under short-day conditions were ground to a fine powder in liquid nitrogen, and then added 50 ml of extraction buffer 1 (0.4 M sucrose, 10 mM Tris-HCl [pH 8.0], 10 mM MgCl2, 5 mM β-ME, 0.1 mM PMSF and 1× Complete protease inhibitor cocktail (Roche)). Extracts were filtered by two layers of Miracloth and then centrifuged at 3000 g for 20 min at 4 °C. The precipitates were lysed with 10 ml of IP buffer

(100 mM Tris-HCl [pH 7.5], 300 mM NaCl, 2 mM EDTA, 1% Triton X-100, 10% glycerol, 1 mM PMSF and 1× Complete protease inhibitor cocktail (Roche)) at 4 °C for 30 min. After centrifugation at 14,000 g at 4 °C for 15 min, the supernatant was diluted by equal volume dilution buffer (100 mM Tris-HCl [pH 7.5], 2 mM EDTA, 10% glycerol, 1 mM PMSF and 1× Complete protease inhibitor cocktail (Roche)), and then incubated with anti-GFP beads (KTSM-life, Cat. No. KTSM1301) at 4 °C for 3 h with gently rotation. Beads were then washed three times with 1×PBS solution containing 0.1% Tween-20. Proteins were eluted in SDS loading buffer and incubated at 55 °C for 10 min, followed by immunoblotting or silver staining.

For mass spectrometry, the immunoprecipitated proteins were eluted using 0.2 M glycine solution (pH 2.5), and then subjected to reduction with dithiothreitol, alkylation with iodoacetamide and digested with trypsin (Thermo Fisher, Cat. No. 90057, MS grade). The high-performance liquid chromatography system Ultimate 3000 UHPLC coupled with Q Exactive HF was used for mass spectrometry analysis. It employs a data-dependent acquisition (DDA) mode for scanning detection, where the top 20 most abundant precursor ions are selected for fragmentation analysis. Mobile phase A consists of a 0.1% formic acid solution, while mobile phase B is an 80% acetonitrile solution containing 0.1% formic acid. The liquid phase system operates at a flow rate of 400 nl/min for the Nano pump and 10 μl/min for the loading pump. The Nano Trap column (Lot#164535, Bed Length = 20 mm) and the analytical column (Lot#164941, length = 250 mm) used are provided by Thermo Fisher Scientific. Two biological replicates were included in the IP-MS analysis. RAW files were processed by Sequest HT algorithm[72], implemented in Thermo Proteome Discoverer (version: 2.4.1.15) software. Tryptic peptides with a minimum length of 6 amino acids, maximum length of 144 amino acids, and 2 maximum missed cleavage sites were searched, with a precursor mass tolerance of 10 ppm and fragment mass tolerance of 0.02 Da. Proteins were then identified with Thermo Proteome Discoverer against the *Arabidopsis thaliana* TAIR10 plus GFP database, following manufactures' instructions with default settings. Spectra from repeated fragmentation of one peptide ion were not combined.

### Immunoblotting

Proteins were loaded onto 4%–20% gradient protein gels (GenScript, SurePAGE, Cat. No. M00655), 4%–20% Precast Protein Plus Gel (Yeasen, Cat. No. 36256ES10) or 10% SDS-PAGE gels at 150 V for 2 h. A wet transformation was performed at 90 V for 90 min in ice-cold transfer buffer. After that, the membranes were blocked in 5% non-fat milk at room temperature for 1 h on a shaking table. Finally, the blocked membranes were incubated in the corresponding antibodies solutions at room temperature for another 3 h. The following antibodies were used: anti-GFP (Abcam, Cat. No. ab290, 1:10 000 dilution), anti-HA (Sigma-Aldrich, Cat. No. H6533, 1:5000), anti-FLAG (Sigma-Aldrich, Cat. No. A8592, 1:5000), anti-H3 (Proteintech, Cat. No. 17168-1-AP, 1:10 000), and horseradish peroxidase-conjugated goat-anti-rabbit secondary antibody (Abcam, Cat. No. ab6721, 1:10 000). The intensities of blotting signals were quantified using ImageJ software (v.1.50i). Uncropped scans of immunoblotting results are shown in Source data.

### GST-pull down assays

For *GST-BCL7A-YFP* and *GST-BCL7B-CFP* construction, the full-length coding sequence of BCL7A or BCL7B (without the stop codon) was fused with yellow florescence protein (YFP) fragment or cyan florescence protein (CFP) fragment by Overlapping PCR. The PCR products were subcloned into the BamHI site of *pGEX-4T-1* vector. For *MBP-BRM (422-952)-HIS* construction, the fragment of BRM (422-952 aa) was amplified from Col-0 cDNA and cloned into the BamHI site of the modified *pMAL-c5x-HIS* vector[73]. MBP-BRM (422-952)-HIS, GST, GST-BCL7A-YFP, and GST-BCL7B-CFP proteins were purified from

transgenic *Escherichia coli* BL21 (DE3) strain as previously described. The GST-pull down assay was performed as follow: 2 μg of purified GST-fusion recombinant proteins and 2 μg of purified MBP-fusion proteins were incubated in 1 ml of binding buffer (50 mM Tris-HCl [pH 7.5], 100 mM NaCl, 0.6% Triton X-100) at 4 °C for 2 h. Glutathione Sepharose 4B beads (GE Healthcare, Cat. No.17075601) were added and incubated at 4 °C for another 2 h. After washing with binding buffer four to six times, precipitated proteins were eluted in SDS loading buffer and boiled at 95 °C for 5 min, followed by SDS-PAGE. The proteins were immunoblotted with anti-GST (TransGen Biotech, Cat. No. HT601-01, 1:5000) or anti-MBP (TransGen Biotech, Cat. No. HT701-1, 1:3000) primary antibodies and subsequently horseradish peroxidase-conjugated goat-anti-mouse secondary antibody (Abcam, Cat. No. ab6789, 1:10 000). The intensities of blotting signals were quantified using ImageJ software. Uncropped scans of immunoblotting results are shown in Source data.

### RNA isolation, RT-qPCR and RNA-seq analyses

Total RNA was extracted from 14-day-old Arabidopsis seedlings grown under short-day conditions using TRIzol reagent (Invitrogen) following the manufacturer's instructions. One microgram of RNA was reverse transcribed using PrimeScript™ RT reagent Kit with gDNA Eraser (Takara, Cat. No. RR047A). For the synthesis of the first cDNA strand of mature miR156, the primers for reverse transcription were designed as previously reported[74]. Reverse transcription program was set as follows: one cycle at 16 °C for 30 min, 60 cycles at 30 °C for 30 s, 42 °C for 30 s, 50 °C for 1 s, and finally one cycle at 85 °C for 5 min to inactivate the reverse transcriptase. The qPCR assays were performed using SYBR Green Supermix (Vazyme, Cat. No. Q711-02) in the StepOne Plus (Applied Biosystems). Results were repeated with three independent RNA samples (biological replicates). Quantification was analyzed with the relative -ΔΔCt method. *ACTIN* and *AtsnoR101* were served as the control for mRNA and miRNA analyses, respectively. The sequences of primers used are listed in Supplementary Data 4.

For RNA-seq analyses, RNAs from three biological replicates were sequenced separately at Novogene, using Illumina Hiseq X-Ten (sequencing method: Hiseq-PE150). The paired-end RNA-seq reads were quantified by Salmon[75] with parameters -k 31 and --type quasi. Salmon index was generated using AtRTD2.fa[76] as the reference transcriptome. The gene quantification files of all samples were used as inputs (TPM values) for differential gene expression analysis by DESeq2[77]. To analyze differential gene expression in all samples, we used "parametric" Fit type, "ratio" method for estimate Size Factors and Wald significance test function provided by DESeq2. The threshold of differential gene expression is "p.adj <0.05 and abs(log$_2$FoldChange) > 0.6". To explore the relationships in transcriptomic changes among different mutants, genes with differential expression in at least one of the seven mutants (*bcl7a*, *bcl7b*, *brm-3*, *brm-1*, *bcl7a bcl7b*, *brm-3 bcl7a bcl7b*, and *brm-1 bcl7a bcl7b*) were collected, resulting in 15,924 genes which were further clustered via hierarchical clustering[78] and principal component analysis (PCA) (https://biit.cs.ut.ee/clustvis/). The hypergeometric test was used to calculate the significance of the overlap of two groups of genes drawn from the set of genes. The total number of genes in the Arabidopsis genome used was 34,218 (27,655 coding and 6563 non-coding genes) according to EnsemblPlants (http://plants.ensembl.org/index.html).

### GUS staining

For GUS staining, 14-day-old Arabidopsis seedlings grown under short-day conditions were collected and pretreated with 90% acetone on ice for 20 min. After washing three times with 100 mM phosphate buffer (pH 7.0), plants were submerged in X-Gluc solution and vacuumed 30 min and then incubated at 37 °C overnight. Stained plants were decolorized in 70% ethanol and photographed.

## ChIP and ChIP-seq analysis

ChIP experiments were performed as previously described[15,31] with minor changes. In brief, 14-day-old seedlings (~0.5 g for each biological replicate) grown on ½-strength MS medium under short-day conditions were fixed using 1% formaldehyde under a vacuum for 15 min and then ground into a fine powder in liquid nitrogen. The chromatin was sonicated into ~300 bp fragments using a Bioruptor sonicator with a 30/30-s on/off cycle (27 total on cycles) at the high setting. Immunoprecipitation was performed using 1 μl of anti-GFP (Abcam, Cat. No. ab290) at 4 °C overnight. ChIP-qPCR was performed with three biological replicates, and the results were calculated as a percentage of input DNA according to the Champion ChIP-qPCR user manual (SABioscience). The primers used for ChIP-qPCR are listed in Supplementary Data 4.

For ChIP-seq, 1 g of seedlings was used, and the ChIPed DNA was purified using the MinElute PCR purification kit (Qiagen, Cat. No. 28004). Libraries were constructed with 1–2 ng of ChIPed DNA using the VAHTS Universal DNA Library Prep Kit for Illumina V3 (Vazyme Biotech, Cat. No. ND607), VAHTSTM DNA Adapters set3–set6 for Illumina (Vazyme Biotech, Cat. No. N805), and VAHTS DNA Clean Beads (Vazyme Biotech, Cat. No. N411-02) according to the manufacturer's protocol. High-throughput sequencing was performed on the Illumina NovaSeq platform (sequencing method: NovaSeq-PE150).

ChIP-seq data analysis was performed as previously described[14,15]. In brief, raw data was trimmed by fastp with following parameters: "-g -q 5 -u 50 -n 15 -l 150". The clean data was mapped to the *A. thaliana* reference genome (TAIR10) using Bowtie 2 with the default settings[79]. Only perfectly and uniquely mapped reads were used for further analysis. A summary of the number of reads for each sample is given in Supplementary Data 5. MACS 2.0[80] was used for peak calling with the following parameters: "gsize = 119,667,750, bw = 300, q = 0.05, nomodel, extsize = 200." The aligned reads were converted to wiggle (wig) formats, and bigwig files were generated by bamCoverage with "-bs 10" and "-normalizeUsing RPKM (reads per kilobase per million)" in deepTools[81]. The data were imported into the Integrative Genomics Viewer (IGV)[82] for visualization. Only peaks that were present in both biological replicates (irreproducible discovery rate ≥0.05) were considered for further analysis. To annotate peaks to genes, ChIPseeker[83] was used with default settings, except that 2 kb of the TSS upstream region and 2 kb of the TSS downstream region were required. Differential occupancy of BRM or BCL7A/B in different mutants was determined using DiffBind[84] with default settings. Venn diagrams were created using Venny (v.2.1) (https://bioinfogp.cnb.csic.es/tools/venny/index.html) to compare overlaps between different groups of genes. ComputeMatrix and plotProfile[81] were used to compare the mean occupancy density of BRM and BCL7A/B at defined loci (details are shown in each corresponding figure legends).

To analyze read density and correlation between different ChIP-seq samples, we performed spearman correlation analysis. Reads density was analyzed over the merged set of binding sites across all ChIPs using multiBigwig-Summary function from deepTools[81] under bin model (bin size = 1000). The heatmap of spearman correlation was generated by PlotCorrelation function in deepTools[81]. Peak overlaps were analyzed by Bedtools intersect function.

For spike-in ChIP-seq, we performed the assay using the *Drosophila* chromatin (Cat. No. 53083) and H2Av antibody (Cat. No. 61686) obtained from the Active Motif company following the manufacturer's instructions. The clean data was mapped to the *Arabidopsis thaliana* reference genome (TAIR10) and *Drosophila melanogaster* (Dm6) using Bowtie 2 with the default settings[79]. The unique mapped reads were then scaled by the reference derived normalization factor to generate RPKM to visualization.

## ATAC-seq experiment and data analyses

About 1 g of 14-day-old Arabidopsis seedlings grown under short-day conditions were collected and cut into small pieces with blade. Then, 5 ml of Enzyme solution (20 mM MES [pH 5.7], 1.5% cellulase R10, 0.4% macerozyme R10, 0.4 M mannitol, 10 mM CaCl2, 3 mM β-mercaptoethanol and 0.1% BSA) were added to the chopped plant tissue to isolate protoplasts as previously described[85]. After counting protoplasts under the microscope using a hemacytometer, about 50,000 protoplasts were used to isolate nuclei by using 5 ml of lysis buffer (1× PBS [pH 7.5], 0.5% Triton X-100 and 1× Complete Protease Inhibitor Cocktail (Roche)). The crude nuclei were washed three times with Nuclei Extraction Buffer (1× PBS [pH 7.5], 0.25 M Sucrose, 1 mM PMSF, 1 mM β-mercaptoethanol, 0.5% Triton X-100, 1× Complete Protease Inhibitor Cocktail (Roche)) and washed once with Tris-Mg buffer (10 mM Tris-HCl [pH 8.0], 5 mM MgCl$_2$). The purified nuclei were then incubated with Tn5 transposome and tagmentation buffer at 37 °C for 30 min (Vazyme Biotech, Cat. No. TD501). After the tagmentation, the DNA was purified using MinElute PCR purification kit (Qiagen, Cat. No. 28004) and then amplified using TruePrepTM DNA Library Prep Kit V2 for Illumina® (Vazyme Biotech, Cat. No. TD501) for 9 cycles. The PCR cycle number was determined as described according to published methods[52]. The index primers were from TruePrepTM Index Kit V2 for Illumina® (Vazyme Biotech, Cat. No. TD202). Amplified libraries were purified with VAHTS DNA Clean Beads (Vazyme Biotech, Cat. No. N411-02). Two biological replicates were performed. High-throughput sequencing was performed on the Illumina NovaSeq platform (sequencing method: NovaSeq-PE150).

ATAC-seq data analyses were performed according to published methods[52] with some modifications. In brief, raw data was trimmed by fastp set adapter sequence with parameter "-a CTGTCTCTTATACA-CATCT". The clean data was mapped to the *A. thaliana* reference genome (TAIR10) using Bowtie 2 with the default settings. A summary of the number of reads for each sample is given in Supplementary Data 2. The online tool Filter BAM datasets was used with parameter "mapping quality ≥30,! Mt,! Pt" to removed unmapped and organelle reads. The online tool MarkDuplicates was used with the default settings to remove duplicated reads. The peak calling, peak annotation, bamCoverage, and IGV visualization were performed according to the methods used for ChIP-seq data analyses. Differential DNA accessibility between mutant and WT (FDR < 0.05, |log$_2$(fold change)| ≥0.5) was determined using DiffBind[84] with default settings. ComputeMatrix and plotProfile[81] were used to compare the mean DNA accessibility density of mutants and WT at defined loci (details are shown in each corresponding figure legends). Correlation between ATAC-seq in different mutants was determined using the deepTools[81]: bigwigCompare, multiBigwigSummary and plotCorrelation with default settings.

## MNase-seq

The MNase assays were carried out as described previously[86] with minor modifications. Approximately 0.4 g of seedlings were used per sample. For MNase treatment, the prepared nuclei were resuspended in prewarmed MNase digestion buffer (20 mM Tris-HCl [pH 8.0], 5 mM NaCl, and 2.5 mM CaCl$_2$), followed by the addition of 0.5 units of MNase (Takara, Cat. No. 2910 A,) and incubation (10 min at 37 °C with periodic agitation). Finally, the mono-nucleosome DNA was isolated from 2% agarose gels and quantified by the Qubit dsDNA HS assay kit (Thermo Fisher, Cat. No. Q32851).

MNase-seq data analyses were performed according to published methods[87] with some modifications. In brief, raw reads were processed with cutadapt (version 3.4, -m 10) to remove adaptors. Secondly, the processed reads were mapped to the *Arabidopsis thaliana* genome (TAIR10) by Bowtie2 with default settings[79]. Thirdly, the nucleosome positions were determined by applying the improved nucleosome-positioning algorithm iNPS with default settings[88]. And then, the

smoothed bigwig files across biological replicates were generated by using 'dpos' function in DANPOS3[89]. Finally, MNase-seq data among samples were normalized by applying quantile normalization methods to normalize occupancy with 'wiq' function in DANPOS3. For Supplementary Fig. 11, we selected the genes with unchanged accessibility in *brm-1* ($n = 1172$) as control.

## Gene ontology analysis

Gene Ontology (GO) enrichment analysis was performed using the DAVID Bioinformatics Resources (https://david.ncifcrf.gov/) and plotted at HIPLOT (hiplot-academic.com).

## Phylogenetic analysis

The amino acid sequences of orthologs proteins of BCL7A/B in animals and plants were downloaded from the NCBI database or UniProt database and used for phylogenetic analysis. The phylogenetic tree was constructed with MEGA11[90] using the neighbor-joining method with 500 bootstrap replicates and the Poisson model. Finally, the phylogenetic tree was beautified using Interactive Tree Of Life[91] (https://itol.embl.de/).

## Statistics and reproducibility

All statistics performed in this manuscript are detailed above, and statistical test methods, sample sizes and *p* values are indicated in the corresponding figure legends. Two-tailed Student's *t* test was conducted using Excel. The *p* values for the Venn diagram overlap analysis are based on hypergeometric tests (http://nemates.org/MA/progs/overlap_stats.html). The *post hoc* Tukey's HSD test was performed in (https://astatsa.com/OneWay_Anova_with_TukeyHSD/). The *r* values of Pearson correlation were calculated using Excel. The *p* values in Figs. 2j, k, 3g, 4h, m, Supplementary Fig. 10e, h, k were calculated using Mann-Whitney U test (https://www.statskingdom.com/170median_mann_whitney.html). All experiments were repeated with at least two biological replicates. No data were excluded from the analyses. The investigators were not blinded to allocation during experiments and outcome assessment.

## Reporting summary

Further information on research design is available in the Nature Portfolio Reporting Summary linked to this article.

## Data availability

The ChIP-seq, MNase-seq, RNA-seq, and ATAC-seq datasets have been deposited in the Gene Expression Omnibus under accession no. GSE215295, GSE215151, and GSE252623. The mass spectrometry proteomics data have been deposited in the iProX database under the dataset identifier IPX0005203001. The BRM, H4K5ac, BRD1, BRD2, and BRD13 ChIP-seq data were downloaded from GEO under accession no. GSE161595. BRIP1 and BRIP2 ChIP-seq data were downloaded from GEO under accession no. GSE142369. The H3K27me3 ChIP-seq data were downloaded from GEO under accession no. GSE145387. The SYD and MINU ChIP-seq data were downloaded from GEO under accession no. GSE218841. The *brm*, *syd*, and *minu* ATAC-seq data used in Supplementary Fig. 10 were downloaded from GEO under accession no. GSE193397. The WT ATAC-seq data used in Supplementary Fig. 8a-e were downloaded from Beijing Institute of Genomics Data Center under BioProject PRJCA002620. Source data are provided with this paper. Uncropped scans of immunoblotting results are shown in Source data. Source data are provided with this paper.

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

## Acknowledgements

We thank the Arabidopsis Biological Resource Center (ABRC) for seeds of T-DNA insertion lines, Prof. Gang Wu (Zhejiang Agriculture and Forestry University) for providing the *spl9-4 spl15-1*, *p35S:MIM156*, and *pMIR156A:GUS* transgenic plants, and Prof. Shi Xiao (Sun Yat-sen University) for providing the *pFGC-UBQ10pro* and *pMAL-c5x-HIS* plasmids. This work was supported by the National Natural Science Foundation of China to C.L. (32070212 and 32270362), to Y.L. (32000249), and to Y.Y. (32200279), China Postdoctoral Science Foundation to Y.Y. (2022M723667), the Guangdong Basic and Applied Basic Research Foundation to C.L. (2021A1515011286) and to Y.L. (2019A1515011216), and the Fundamental Research Funds for the Central Universities to C.L. (18lgzd12) and to Y.L. (19lgpy196).

## Author contributions

C.L. conceived the project. Y.L. and Y.Y. performed most of the experiments. Y.L., Y.Y., W.F. and T.Z. conducted bioinformatics analysis. Y.L., Y.Y., C.W., Z.Z., Z.Y., X.S., J.X., Z.L., P.L. and C.L. analyzed data. C.L. wrote the manuscript.

## Competing interests

The authors declare no competing interests.

## Additional information

Yawen Lei [1,3], Yaoguang Yu[1,3], Wei Fu[1], Tao Zhu[1], Caihong Wu[1], Zhihao Zhang[1], Zewang Yu[1], Xin Song[1], Jianqu Xu[1], Zhenwei Liang[1], Peitao Lü [2] & Chenlong Li [1] ✉

[1]State Key Laboratory of Biocontrol and Guangdong Key Laboratory of Plant Resources, School of Life Sciences, Sun Yat-sen University, Guangzhou 510275, China. [2]College of Horticulture, FAFU-UCR Joint Center for Horticultural Biology and Metabolomics, Haixia Institute of Science and Technology, Fujian Agriculture and Forestry University, Fuzhou 350002, China. [3]These authors contributed equally: Yawen Lei, Yaoguang Yu. ✉e-mail: lichlong3@mail.sysu.edu.cn

