## [Peer Review File · Nature Communications]

REVIEWER COMMENTS

Reviewer #1 (Remarks to the Author):

The manuscript by Yawen Lei explores the role of BCL7 proteins A and B in SWI/SNF complex functions. The authors propose that BCL7 are components of the BRM containing SWI/SNF complex and that their incorporation into the complex requires BRM but they are not required for SWI/SNF complex assembly or chromatin targeting. In contrast, the authors propose that BCL7 proteins potentiate SWI/SNF complex ability to remodel nucleosomes.

This is an interesting conclusion that offers a fresh look at the SWI/SNF complex biology and one that in my opinion is worth pursuing.

In my opinion, most of the work is done properly with proper controls. Below I list problems I like the authors to address in the order they appear in the text:

1. Figure 1d What is BRM-N-GFPD ?

2. line 204 "Consistently, heatmap analysis showed that BCL7A, BCL7B, BRIP1/2, BRD1/2/13, and H4K5ac but not H3K27me3, all exhibited significant enrichments at BRM binding peak summits (Fig. 1f). "

no statistics were included so the statement about "significance" is not relevant here.

3. Are the positions where BCL7 is lost in brm-1 enriched for the BRM/BCL7 common binding sites?

4. Are the brm-1 unaffected BCL7 binding sites enriched in SYD and MINU binding sites?

5. The overlap between ATACseq and RNAseq affected genes in BCL7 mutant is very modest. How would the overlap look for BRM in brm-1 ATACseq/RNAseq?

This questions the mode of BCL7 activity so should be addressed in details.

6. Fig 6e changes in ATAC seq signal in bcl7 are quite mild in contrast to changes observed in ATAC signal in brm-1.

7. ATACseq signal in bcl7ab is reduced to a similar extent in retained and reduced sites (Fig.4c) Why?

Finally, SNF2 proteins like BRM can have ATP-dependent and independent functions. The model proposed by the authors suggests that BCL7 do not affect BRM binding to chromatin but nucleosome remodelling activity presumably by stimulating its ATPase activity.

In my opinion, the manuscript would be immensely strengthened by providing the in vitro ATPase activity, and remodelling activity of BRM in the presence or absence of BCL7.

If the authors are right and BCL7 somehow stimulates the BRM ability to remodel nucleosomes one can expect that the effect of BCL7 on ATPase dependent and ATPindependent BRM functions will be different. Can this be tested?

Reviewer #2 (Remarks to the Author):

In this work, the authors studied the roles of BCL7A/B in regulating BRM activity by using a bunch of omics and biochemical approaches. They found that BCL7A/B physically interact with the key component of BAS complex, BRM, which is required for BCL7A/B to associate with BAS complex. CHIP-seq data show that BCL7A/B and BRM share a substantial proportion of genomic targets, which are mainly around TSSs of the target genes. Interestingly, their data demonstrate that BCL7A/B act to regulate the remodeler activity but not the assembly or genomic targeting of the BAS complex. Since BCL7A/B are the conserved components of BAS complex in plants and animals, the findings of this work suggest a mechanism of BCL7A/B regulating BAS complex in animals. The authors further identified MIR156A/C as a target of BAS complex, and showed that BCL7A/B regulates chromatin accessibility and expression but not BAS targeting of MIR156A/C, which is well consistent with the genomic data and can explain how BCL7A/B-containing BAS complex regulates vegetative phase transition. This work uses elegant techniques and bioinformatic analyses to study how a chromatin remodeler complex is fine-tuned, which is of broad interest for scientists in epigenetics community. Experiments are well designed, results are well explained, and conclusions are well supported by the results.

Specific minor concerns are as follows:

1. The MNase-seq data was not well explained. The data clearly showed that nucleosome occupancy on the target genes is increased in *brm* and *bcl7* mutants compared to WT (Extended Data Fig. 10a-b). It is known that nucleosomes have inhibitory effects on chromatin accessibility and gene expression. Therefore, their MNase-seq data suggest a mechanism for BAS complex regulating chromatin accessibility, that is, the BAS complex removes or loosens nucleosomes from chromatin to maintain chromatin accessibility. The authors could suggest such a mechanism in the manuscript. In addition, the authors should include a control analysis with BRM-unbound genes or the genes with unaltered chromatin accessibility in the mutants to show unchanged nucleosome occupancy on them between WT and the mutants.
2. The authors showed reduced chromatin accessibility and expression of MIR156A/C loci, which is nicely in line with SPL upregulation in the mutants. To rule out the alternative possibility that BAS complex might directly regulate SPLs bypassing miR156, the authors could show ATAC-seq and CHIP-seq data for SPL loci in WT and the *brm* and *bcl7* mutants. In addition, the authors could do more thorough bioinformatic analysis on all the genes regulated by BCL7A/B and BRM and indicate whether other genes/pathways might be possibly involved in vegetative phase transition than MIR156-SPL pathway.
3. Fig. 2l, m are mislabeled and should be BCL7A/B ChIPseq instead of BRIPs according to the text. Please clarify.
4. Although discussed, it is not clear to me how BAS complex might maintain chromatin accessibility of MIR156A/C from embryonic stage to post-embryonic stage. They mentioned that BAS complex exists throughout the life cycle of plants, however, MIR156A/C expression goes through phases. How does BAS complex maintain MIR156A/C expression in juvenile stage but doesn't do so in adult stage? The authors may explain more in the Discussion and revise the abstract accordingly.

Reviewer #3 (Remarks to the Author):

I was asked by the editor to focus on the IP-MS in this manuscript.

The results presented in the manuscript are interesting and significant for describing the mechanisms of chromatin remodeling in plants. Proteomic experiments led to interesting conclusions but there are few flaws regarding description or interpretation of these results. The methods of mass spectrometric analyses are not well described.

Results

The first problem is no negative controls in protein identification results presented on Ext. Fig 1a; Fig. 2abc and Ext. Fig 5b. The proteins tend to stick to beads used for IP-MS experiments which could lead to false positive identifications. The amount of unspecific binding is often very sensitive to many, difficult to control, factors during the experiment. The easiest way to deal with this problem is negative control in each experiment. For example, when using GFP fusion with bait protein, the easiest way is to make IP-MS on wild type plants as negative control. I suppose that authors made such negative controls. These data should be presented. We do not know if some peptides of proteins from SWI/SNF complex were identified in negative controls.

On the Fig. 2s and Ext. Fig 5b one can see that SWI3C, BRIP1/2, BRD1/2/13 do not identify in IP-MS of brm-1 mutant. This allows us to draw conclusions about these proteins without negative control. In the case of SWP73B, ARP4/7, ACT2/7 and GIF2 we cannot exclude unspecific binding to the beads. Therefore we can not be sure if they bind to BRM or to the resin. Authors show that there is significantly less ARP4/7 peptides in brm plants. Maybe only a fraction of ARPs binds to the complex and the rest is identified because of unspecific binding? Showing negative controls would solve that problem.

Fig. 2bc – I would be very cautious when determining significance only on the base of identification of one or two peptides (case of BRIP1). When performing spectral counting for comparing the amounts of proteins it is essential to identify at list few peptides. Quite often the identification score of peptide in near the significance threshold making the peptide randomly appears in some samples and disappears in others. The scores of peptides are not shown in tables therefore I could not see if the peptides are identified with scores near threshold. There is more proteins identified with one or two peptides (ACT2/7, GIF2). To conclude on the base of identification of these peptides one should check the scores of peptides and screen the list of peptides identified below the threshold in all samples.

Methods

Authors write that Thermo Scientific Q Exactive HF mass spectrometer was used in data-dependent mode (line 722). There is no information about chromatographic system used (nanoHPLC/UPLC or EvoSep, column type and size, gradient type and length) and mass spectrometer settings.

We can find the information that spectral data were searched against the TAIR10 database using Protein Prospector 4.0 (line 723) but the search parameters are not described. After that authors write that RAW data were searched against TAIR10 database with MaxQuant software. Which software was used?

Then authors write that Label-free quantitation (LFQ) in MaxQuant was performed. LFQ is peak intensity-based quantitative method but there are only numbers of identified peptides presented on figures. There are no LFQ data presented.

The description of mass spectrometric data analyses does not allow to determine how the samples were analysed.

I could not find data under identifier IPX0005203000 in the Integrated Proteome Resources (<https://www.iprox.cn/>).

Response to the comments of Reviewers:

Reviewer #1 (Remarks to the Author):

The manuscript by Yawen Lei explores the role of BCL7 proteins A and B in SWI/SNF complex functions. The authors propose that BCL7 are components of the BRM containing SWI/SNF complex and that their incorporation into the complex requires BRM but they are not required for SWI/SNF complex assembly or chromatin targeting. In contrast, the authors propose that BCL7 proteins potentiate SWI/SNF complex ability to remodel nucleosomes.

This is an interesting conclusion that offers a fresh look at the SWI/SNF complex biology and one that in my opinion is worth pursuing.

In my opinion, most of the work is done properly with proper controls. Below I list problems I like the authors to address in the order they appear in the text:

We appreciate reviewer 1 for the very positive feedback and many insightful suggestions.

1. Figure 1d What is BRM-N-GFPD?

Response: The “BRM-N-GFPD” should be “BRM-N-GFP”, and we have corrected it in the revised manuscript (see Figure 1b).

2. line 204 "Consistently, heatmap analysis showed that BCL7A, BCL7B, BRIP1/2, BRD1/2/13, and H4K5ac but not H3K27me3, all exhibited significant enrichments at BRM binding peak summits (Fig. 1f). "

no statistics were included so the statement about "significance" is not relevant here.

Response: Thanks for the comments. We have replaced the "significance" with “substantial”:

“Consistently, heatmap analysis showed that BCL7A, BCL7B, BRIP1/2, BRD1/2/13, and H4K5ac but not H3K27me3, all exhibited substantial enrichments at BRM binding peak summits (Fig. 1f).”

3. Are the positions where BCL7 is lost in brm-1 enriched for the BRM/BCL7 common binding sites?

Response: Thanks for the question. Yes, we found that the genomic sites where

BCL7A/B were lost in *brm-1* showed a significant overlap with BRM and BCL7 co-target sites, indicating that the localization of BCL7A/B on many target genes is dependent on the direct presence of BRM ATPase. The new data have been included in the revised Extended Data Fig. 6a-b and described in lines 250-253. Thanks again.

Response Figure 1 | Venn diagrams displaying statistically significant overlaps between the genes where BCL7A or BCL7B binding signal reduced in *brm-1* and those co-targeted by BRM/BCL7.

4. Are the *brm-1* unaffected BCL7 binding sites enriched in SYD and MINU binding sites?

Response: Thanks for your question. Yes, the BCL7 binding sites unaffected in *brm-1* were enriched in both SYD and MINU binding sites, implying that the binding of BCLs at those sites may be directly regulated by SYD and/or MINU. The new data have been included in the revised Extended Data Fig. 6c-f and described in lines 253-255.

Response Figure 2 | Venn diagrams displaying statistically significant overlaps between the genes where BCL7A or BCL7B binding signal unaffected in *brm-1* and those target by SYD/MINU.

5. The overlap between ATACseq and RNAseq affected genes in BCL7 mutant is very modest. How would the overlap look for BRM in *brm-1* ATACseq/RNAseq?

This questions the mode of BCL7 activity so should be addressed in details.

Response: Thank you for pointing this out. In our study, we observed a relatively mild overlap between the misregulated genes identified by RNA-seq and ATAC-seq in *bcl7ab* mutants. The overlap of down-regulated genes between ATAC-seq and RNA-seq in *brm-1* mutants was also modest and comparable to that observed in *bcl7a bcl7b* mutants (Representation factor is 1.6 for *brm* mutants and 1.4 for *bcl7* mutants) (see Response Figure 3). The mild yet significant overlapping observed in *bcl7* mutants might be explained by two reasons. First, it is generally accepted that chromatin accessibility change is necessary, but not sufficient, for changes in gene transcription based on evidence from various systems. Current evidence in the field argues for a

multilayered regulatory programme in which chromatin accessibility plays a broad licensing role but does not determine the gene expression state (Klemm et al., 2019; Wang et al., 2020). Second, the loss of BCL7A/B compromises but does not completely destroy the BAS-mediated chromatin accessibility changes. Therefore, the mild changes in accessibility caused by the loss of BCL7 might account for the mild changes in gene expression. BRM, in contrast, serves as the core ATPase within the BAS complexes; its absence should completely destroy the remodeling activity, leading to a more pronounced impairment of chromatin accessibility and gene transcription compared to the loss of BCL7. We have discussed these two possibilities in the revised manuscript to better interpret our data (lines 554-568). We appreciate the reviewer's understanding and hope you are satisfied by our explanation.

Response Figure 3| The overlap of down-regulated genes between ATAC-seq and RNA-seq in *brm-1* and *bcl7a bcl7b*.

6. Fig 6e changes in ATAC seq signal in *bcl7* are quite mild in contrast to changes observed in ATAC signal in *brm-1*.

Response: Thank you for the comment. Indeed, the loss of BCL7A/B compromises but does not completely destroy BAS remodeling activity. Therefore, BCL7 subunits are required for fine-tuning the SWI/SNF activity at target genes for optimized gene transcription regulation. BRM, in contrast, serves as the central ATPase within the BAS complexes; its absence should completely destroy the remodeling activity, leading to a more pronounced impairment of chromatin accessibility and gene transcription compared to the loss of BCL7. We thanks you are satisfied by our explanation.

7. ATACseq signal in *bcl7ab* is reduced to a similar extent in retained and reduced sites (Fig.4c) Why?

Response: Thank you for pointing this interesting question out. The retained sites in *brm* mutants might be regulated by MAS-, SAS-subcomplexes, or jointly by all three subcomplexes (MAS, SAS, and BAS). Because BCL7A/B are also subunits of MAS and SAS, they may regulate accessibility at these *brm*_retained sites, which could explain the observed reduction of ATAC signal in *bcl7ab* mutants. Indeed, among the 6,250 *brm*_retained sites, 1,184 and 472 sites showed a significant reduction in accessibility in SAS mutants and MAS mutants, respectively (Response Figure 4a,c,e). They also displayed reduced accessibility in the *bcl7ab* mutants (Response Figure 4b,d). The remaining 4,519 sites may be redundantly regulated by two or three subcomplexes. Thus, the lack of one subcomplex may not cause a reduction in accessibility (Response Figure 4g,h), but lack of BCL7A/B may affect the activity of all three subcomplexes, accounting for the reduced accessibility in the *bcl7ab* mutants (Response Figure 4f). Thanks again for the question.

Response Figure 4 | BCL7A/B are likely involved in regulating the accessibility maintained by the SYD- and MINU-complexes.

Finally, SNF2 proteins like BRM can have ATP-dependent and independent functions. The model proposed by the authors suggests that BCL7 do not affect BRM binding to chromatin but nucleosome remodelling activity presumably by stimulating its ATPase activity.

In my opinion, the manuscript would be immensely strengthened by providing the *in vitro* ATPase activity, and remodelling activity of BRM in the presence or absence of BCL7.

If the authors are right and BCL7 somehow stimulates the BRM ability to remodel nucleosomes one can expect that the effect of BCL7 on ATPase dependent and ATP independent BRM functions will be different. Can this be tested?

Response: Thank you for the suggestion on testing the remodeling activity *in vitro*. We

actually have tried many times to purify the intact and active endogenous SWI/SNF complexes from plant tissues for the in vitro assay, but without success so far. Protein complex purification from plant samples remains an extremely challenging task, particularly for big complexes like BAS complex which is ~740 kDa composed of 9 subunits. Another potential alternative system is to individually express the nine subunits of the BAS complex including the BRM ATPase in vitro, and then reassemble them into a complete complex. Ideally, the recombinant expression should be performed using eukaryotic cells such as human or insect cells to make sure the subunits are in their active forms. Unfortunately, we have been unsuccessful in expressing the recombinant ATPase BRM in vitro, possibly due to its high molecular weight of ~250 kDa.

Nevertheless, we realize that although our combined omics experiments (ATAC-seq and ChIP-seq) have provided strong evidence supporting the function of BCL7A/B in fine-tuning BAS activity, in vitro remodeling activity experiments could provide further insights into the function of BCL7s. We would like to explore this more thoroughly in the future when techniques become available on our hands. In light of the reviewer's comment, we have discussed the necessity to perform the in vitro assay to better understand the role of BCL7 in SWI/SNF function (lines 549-553). We appreciate the reviewer's understanding and hope you are satisfied with our explanation.

Reviewer #2 (Remarks to the Author):

In this work, the authors studied the roles of BCL7A/B in regulating BRM activity by using a bunch of omics and biochemical approaches. They found that BCL7A/B physically interact with the key component of BAS complex, BRM, which is required for BCL7A/B to associate with BAS complex. ChIP-seq data show that BCL7A/B and BRM share a substantial proportion of genomic targets, which are mainly around TSSs of the target genes. Interestingly, their data demonstrate that BCL7A/B act to regulate the remodeler activity but not the assembly or genomic targeting of the BAS complex. Since BCL7A/B are the conserved components of BAS complex in plants and animals, the findings of this work suggest a mechanism of BCL7A/B regulating BAS complex in animals. The authors further identified MIR156A/C as a target of BAS complex, and showed that BCL7A/B regulates chromatin accessibility and expression but not BAS targeting of MIR156A/C, which is well consistent with the genomic data and can explain how BCL7A/B-containing BAS complex regulates vegetative phase transition. This work uses elegant techniques and bioinformatic analyses to study how a chromatin remodeler complex is fine-tuned, which is of broad interest for scientists in epigenetics community. Experiments are well designed, results are well explained, and conclusions are well supported by the results.

Thank you very much for the very positive comments.

Specific minor concerns are as follows:

1. The MNase-seq data was not well explained. The data clearly showed that

nucleosome occupancy on the target genes is increased in *brm* and *bcl7* mutants compared to WT (Extended Data Fig. 10a-b). It is known that nucleosomes have inhibitory effects on chromatin accessibility and gene expression. Therefore, their MNase-seq data suggest a mechanism for BAS complex regulating chromatin accessibility, that is, the BAS complex removes or loosens nucleosomes from chromatin to maintain chromatin accessibility. The authors could suggest such a mechanism in the manuscript. In addition, the authors should include a control analysis with BRM-unbound genes or the genes with unaltered chromatin accessibility in the mutants to show unchanged nucleosome occupancy on them between WT and the mutants.

Response: Thank you for the nice suggestion which we totally agreed with. We have now revised the related statements (lines 365-368) as follow: “It is known that nucleosomes have inhibitory effects on chromatin accessibility and gene expression. Therefore, the MNase-seq data also suggest that the BAS complex may remove or loosen nucleosomes from chromatin to maintain chromatin accessibility.”. In addition, the control analysis was performed, and the data were included in revised Extended Data Fig. 11, which showed that the nucleosome occupancy at the genes with unaltered chromatin accessibility was largely unchanged compared to the genes with reduced chromatin accessibility (a minor increase of nucleosome occupancy could be explained by the fact that some of the accessibility retained sites ($|\log_2(\text{fold change})| \geq 0.5$) actually showed a slight reduction of accessibility (see Figure 4b and d)). Thanks again for the excellent suggestion.

2. The authors showed reduced chromatin accessibility and expression of MIR156A/C loci, which is nicely in line with SPL upregulation in the mutants. To rule out the alternative possibility that BAS complex might directly regulate SPLs bypassing miR156, the authors could show ATAC-seq and ChIP-seq data for SPL loci in WT and the *brm* and *bcl7* mutants. In addition, the authors could do more thorough bioinformatic analysis on all the genes regulated by BCL7A/B and BRM and indicate whether other genes/pathways might be possibly involved in vegetative phase transition than MIR156-SPL pathway.

Response: Thank you for the comment. As suggested, we included the ATAC-seq and ChIP-seq data for *SPL9* and *SPL15*, the two major players in regulating vegetative phase transition (Xu et al., 2016b). We observed that BRM and BCL7A/B bind to *SPL9* and *15* (Extended Data Fig. 16). ATAC-seq data showed a decrease in chromatin accessibility of *SPL9/15* in both *brm* and *bcl7a/b* mutants compared to wild-type (Extended Data Fig. 16). However, albeit decreased chromatin accessibility at *SPL9/15*, the mRNA levels of *SPL9/15* are actually increased in *brm* and *bcl7a/b* mutants (Figure 6d). This increase in mRNA levels of *SPL9/15* should be because of the reduced *miR156* levels. Therefore, our data do not support the hypothesis that BAS complex directly regulates SPLs bypassing *miR156*. We have added these data in the revised manuscript and discussed thoroughly (revised Extended Data Fig. 16 and described in lines 615-623). Thanks! (note: The *SPL13* locus did not exhibit detectable ATAC signals. We examined published ATAC-seq data from WT seedlings (Guo, et al., 2022; Wang, et al., 2020) and found that their ATAC-seq also did not detect signals at *SPL13*

locus (Response Figure 5b). Because the reason for the absence of ATAC signals at the *SPL13* gene is currently unknown, we have decided to not include the data for *SPL13* in the revised manuscript).

For your second suggestion, our GO term analysis on the genes regulated by *BCL7A/B* and *BRM* showed that various pathways were enriched, such as response to light stimulus, response to auxin, shoot development, leaf development, and so on (Fig.5i-j, Extended Data Fig. 9). However, these GO pathways have a considerable number of genes and complex regulatory networks, making it difficult to determine which pathways or genes may directly participate in BAS-mediated vegetative phase transition. It would be interesting to explore further in future.

Response Figure 5 | IGV views of ATAC-seq and ChIP-seq signals on *SPL9*, *SPL13*, and *SPL15* in the indicated plant materials.

3. Fig. 2l, m are mislabeled and should be *BCL7A/B* ChIPseq instead of BRIPs according to the text. Please clarify.

Response: Thank you for pointing this error out, and we have corrected in the revised manuscript. Thanks!

4. Although discussed, it is not clear to me how BAS complex might maintain chromatin accessibility of *MIR156A/C* from embryonic stage to post-embryonic stage.

They mentioned that BAS complex exists throughout the life cycle of plants, however, MIR156A/C expression goes through phases. How does BAS complex maintain MIR156A/C expression in juvenile stage but doesn't do so in adult stage? The authors may explain more in the Discussion and revise the abstract accordingly.

Response: Thank you for the question. We have discussed this point in the revised manuscript as follow: "On the other hand, MIR156A/C expression declines through adult phases. How does the BAS complex maintain *MIR156A/C* expression in the juvenile stage but does not do so in the adult stage? Previous studies showed that POLYCOMB REPRESSIVE COMPLEX2 (PRC2) represses *MIR156A/C* through depositing repressive histone mark H3K27me3 (Xu et al., 2016a; Cheng et al., 2021; Gao et al., 2022). The decrease of *miR156* expression with age is temporally correlated with a gradual increase in the level of H3K27me3 and the accumulation of PRC2 complex at the *MIR156A/C* loci following plant growth. It is known that BRM and PRC2 act antagonistically at the nucleosome level to regulate gene expression in plants (Wu et al., 2012; Li et al., 2015). Therefore, the progressive increase of PRC2 complex and H3K27me3 at the *MIR156A/C* loci with age may result in a closed nucleosome status that antagonizes the activity of BAS complex in the adult stage." See lines 603614.

Reviewer #3 (Remarks to the Author):

I was asked by the editor to focus on the IP-MS in this manuscript.

The results presented in the manuscript are interesting and significant for describing the mechanisms of chromatin remodeling in plants. Proteomic experiments led to interesting conclusions but there are few flaws regarding description or interpretation of these results. The methods of mass spectrometric analyses are not well described.

We highly appreciate reviewer 3 for his/her careful evaluation over the IP-MS method.

Results

The first problem is no negative controls in protein identification results presented on Ext. Fig 1a; Fig. 2abc and Ext. Fig 5b. The proteins tend to stick to beads used for IP-MS experiments which could lead to false positive identifications. The amount of unspecific binding is often very sensitive to many, difficult to control, factors during the experiment. The easiest way to deal with this problem is negative control in each experiment. For example, when using GFP fusion with bait protein, the easiest way is to make IP-MS on wild type plants as negative control. I suppose that authors made such negative controls. These data should be presented. We do not know if some peptides of proteins from SWI/SNF complex were identified in negative controls.

Response: Thank you for the comment. Yes, when we were performing the IP-MS assays, we had included the *35S:GFP* or wild-type as negative controls, which showed that peptides of proteins from SWI/SNF complexes were not identified in negative controls (Response Figure 6). We apologize for not showing such a negative control in our previous version of the manuscript. Following your suggestion, we have now

presented these data in the revised manuscript (see revised Extended Data Fig 1a and Extended Data Fig 5d for details). Thanks again!

a				b		
IP: BRM-GFP + mass spectrometry data from published data (Li .et al 2016)				protein	WT	
Protein	Bio Repeats	Number of Unique peptide	Peptide Coverage (%)	IP1	IP2	
BCL7A	IP1	6	21.5	0	0	
	IP2	6	32.91	0	0	
BCL7B	IP1	22	78.0	0	0	
	IP2	14	73.2	0	0	
IP: 35S:GFP + mass spectrometry data from published data (Li .et al 2016)				BRM	0	
Protein	Bio Repeats	Number of Unique peptide	Peptide Coverage (%)	SWI3C	0	
BCL7A	IP1	0	0	BRIP1	0	
	IP2	0	0	BRIP2	0	
BCL7B	IP1	0	0	BRD1	0	
	IP2	0	0	BRD2	0	
				BRD13	0	
				SWP73B	0	
				ARP4	0	
				ARP7	0	
				ACT2	1	
				ACT7	0	
				GIF2	0	

Response Figure 6 | Summary of the peptides of indicated proteins identified by mass spectrometry from the *35S.GFP* line or WT.

On the Fig. 2s and Ext. Fig 5b one can see that SWI3C, BRIP1/2, BRD1/2/13 do not identify in IP-MS of *brm-1* mutant. This allows us to draw conclusions about these proteins without negative control. In the case of SWP73B, ARP4/7, ACT2/7 and GIF2 we cannot exclude unspecific binding to the beads. Therefore we can not be sure if they bind to BRM or to the resin. Authors show that there is significantly less ARP4/7 peptides in *brm* plants. Maybe only a fraction of ARPs binds to the complex and the rest is identified because of unspecific binding? Showing negative controls would solve that problem.

Response: Thank you for the comment. We have added the negative control data from WT or *35S.GFP* samples, which showed that the SWP73B and ARP4/7 proteins are from the BAS complex but not unspecific binding to the beads. (see revised Extended Data Fig 1a and Extended Data Fig 5d for details).

Fig. 2bc – I would be very cautious when determining significance only on the base of identification of one or two peptides (case of BRIP1). When performing spectral counting for comparing the amounts of proteins it is essential to identify at list few peptides. Quite often the identification score of peptide in near the significance threshold making the peptide randomly appears in some samples and disappears in others. The scores of peptides are not shown in tables therefore I could not see if the peptides are identified with scores near threshold. There is more proteins identified with one or two peptides (ACT2/7, GIF2). To conclude on the base of identification of these peptides one should check the scores of peptides and screen the list of peptides identified below the threshold in all samples.

Response: Thank you for the suggestion. We have now presented the scores of peptides in the revised Supplementary Table 1. For Figure 2c, we have decided to delete the ACT2/7, BRIP1, and GIF2 proteins when determining significance, because of their low number (or scores) of peptides (revised Fig.2c). Accordingly, we have now delete BRIP1 and revised the statement as follow: “However, in the absence of BRM, neither

BCL7A nor BCL7B was able to immunoprecipitate SWI3C, BRIP2, and BRD1/2/13 (Fig. 2a-c and Extended Data Fig. 5a-d). Hence, these data imply that BCL7A/B require BRM for incorporating/assembling with the signature subunits to form the BAS complexes.” Thanks again!

	BCL7B-GFP				BCL7B-GFP brm-1			
	IP1		IP2		IP1		IP2	
	Sum PEP Score	Unique Peptides	Sum PEP Score	Unique Peptides	Sum PEP Score	Unique Peptides	Sum PEP Score	Unique Peptides
BCL7B(bait)	41.53	6	36.653	5	38.138	5	37.884	6
BRM	61.532	19	45.688	12	NA	0	NA	0
SWI3C	66.289	16	61.259	13	NA	0	NA	0
BRIP1	3.695	1	6.71	2	NA	0	NA	0
BRIP2	39.933	8	43.91	8	NA	0	NA	0
BRD1	80.444	20	73.396	17	NA	0	NA	0
BRD2	27.943	10	18.546	8	NA	0	NA	0
BRD13	24.037	8	20.353	8	NA	0	NA	0
SWP73B	45.287	11	69.238	14	74.15	16	59.464	14
ARP4	95.354	20	149.017	22	88.784	16	112.474	15
ARP7	83.489	17	109.138	16	73.517	14	75.231	13
ACT2	7.746	1	9.709	2	18.004	2	11.826	1
ACT7	7.02	1	8.994	1	12.924	1	15.958	2
GIF2	4.565	1	7.14	1	7.291	1	NA	0

Response Figure 7 | Summary of sum peptide scores of the specific proteins of BCL7B in WT and *brm-1* background.

Methods

Authors write that Thermo Scientific Q Exactive HF mass spectrometer was used in data-dependent mode (line 722). There is no information about chromatographic system used (nanoHPLC/UPLC or EvoSep, column type and size, gradient type and length) and mass spectrometer settings.

Response: Thanks for pointing this out. We have included information about the chromatographic system used in the methods section (lines 767-775): “The high-performance liquid chromatography system Ultimate 3000 UHPLC coupled with the QExactive HF was used for mass spectrometry data analysis. It employs a data-dependent acquisition (DDA) mode for scanning detection, where the top 20 most abundant precursor ions are selected for fragmentation analysis. Mobile phase A consists of a 0.1% formic acid solution, while mobile phase B is an 80% acetonitrile solution containing 0.1% formic acid. The liquid phase system operates at a flow rate of 400 nl/min for the Nano pump and 10 ml/min for the loading pump. The Nano Trap column (Lot#164535, Bed length = 20 mm) and the analytical column (Lot#164941, length = 250 mm) used are provided by Thermo Fisher Scientific (China)”. The flow gradient setting is as follows:

No.	Time	Flow [μl/min]	% B	Curve
1	0.000		Run	
2	0.000	0.300	3.0	5
3	3.000	0.300	3.0	5
4	7.000	0.300	8.0	5
5	72.000	0.300	36.0	5
6	80.000	0.300	44.0	5
7	85.000	0.300	99.0	5

8	95.000	0.300	99.0	5
9	95.100	0.300	3.0	5
10	105.000	0.300	3.0	5
11	NEW ROW			
12	105.000		Stop Run	

We can find the information that spectral data were searched against the TAIR10 database using Protein Prospector 4.0 (line 723) but the search parameters are not described. After that authors write that RAW data were searched against TAIR10 database with MaxQuant software. Which software was used?

Response: Thank you for the question. We are sorry for the confusion here. The default search parameters of Protein Prospector 4.0 software were used (lines 775-780, see below for details). We did not use the MaxQuant software (it was a mistake to write this). We have now corrected it.

1. Input Data	
Protein Database	AthplusGFP.fasta
Enzyme Name	Trypsin (Full)
Max. Missed Cleavage Sites	2
Min. Peptide Length	6
Max. Peptide Length	144
2. Tolerances	
Precursor Mass Tolerance	10 ppm
Fragment Mass Tolerance	0.02 Da
Use Average Precursor Mass	False
Use Average Fragment Mass	False
3. Spectrum Matching	
Use Neutral Loss a Ions	True
Use Neutral Loss b Ions	True
Use Neutral Loss y Ions	True
Use Flanking Ions	True
Weight of a Ions	0
Weight of b Ions	1
Weight of c Ions	0
Weight of x Ions	0
Weight of y Ions	1
Weight of z Ions	0
4. Dynamic Modifications	
Max. Equal Modifications Per Peptide	3
1. Dynamic Modification	Oxidation / +15.995 Da (M)
2. Dynamic Modification	None
3. Dynamic Modification	None
4. Dynamic Modification	None
5. Dynamic Modification	None
6. Dynamic Modification	None
5. Dynamic Modifications (peptide terminus)	
1. N-Terminal Modification	None
2. N-Terminal Modification	None
3. N-Terminal Modification	None
1. C-Terminal Modification	None
2. C-Terminal Modification	None
3. C-Terminal Modification	None
6. Dynamic Modifications (protein terminus)	
1. N-Terminal Modification	Acetyl / +42.011 Da (N-Terminus)
2. N-Terminal Modification	None
3. N-Terminal Modification	None
1. C-Terminal Modification	None
2. C-Terminal Modification	None
3. C-Terminal Modification	None
7. Static Modifications	
Peptide N-Terminus	None
Peptide C-Terminus	None
1. Static Modification	Carbamidomethyl / +57.021 Da (C)
2. Static Modification	None
3. Static Modification	None
4. Static Modification	None
5. Static Modification	None
6. Static Modification	None

Then authors write that Label-free quantitation (LFQ) in MaxQuant was performed. LFQ is peak intensity-based quantitative method but there are only numbers of identified peptides presented on figures. There are no LFQ data presented.

The description of mass spectrometric data analyses does not allow to determine how the samples were analysed.

Response: We are sorry for the confusion here. The Label-free quantitation (LFQ) analysis was not performed in this manuscript. We mistakenly wrote this. We have now deleted this in the methods.

I could not find data under identifier IPX0005203000 in the Integrated Proteome Resources (<https://www.iprox.cn/>).

Response: Thank you for pointing this out. The data can now be viewed via URL:<https://www.iprox.cn/page/SSV024.html?url=1696754351230fA7E>; Password: fnmY

References

- Cheng, Y.-J., Shang, G.-D., Xu, Z.-G., Yu, S., Wu, L.-Y., Zhai, D., Tian, S.-L., Gao, J., Wang, L., and Wang, J.-W. (2021). Cell division in the shoot apical meristem is a trigger for miR156 decline and vegetative phase transition in Arabidopsis. *Proc Natl Acad Sci U S A* **118**.
- Gao, J., Zhang, K., Cheng, Y.J., Yu, S., Shang, G.D., Wang, F.X., Wu, L.Y., Xu, Z.G., Mai, Y.X., Zhao, X.Y., Zhai, D., Lian, H., and Wang, J.W. (2022). A robust mechanism for resetting juvenility during each generation in Arabidopsis. *Nat Plants* **8**, 257-268.
- Guo, J., Cai, G., Li, Y.Q., Zhang, Y.X., Su, Y.N., Yuan, D.Y., Zhang, Z.C., Liu, Z.Z., Cai, X.W., Guo, J., Li, L., Chen, S., and He, X.J. (2022). Comprehensive characterization of three classes of Arabidopsis SWI/SNF chromatin remodelling complexes. *Nat Plants* **8**, 1423-1439.
- Klemm, S.L., Shipony, Z., and Greenleaf, W.J. (2019). Chromatin accessibility and the regulatory epigenome. *Nat Rev Genet* **20**, 207-220.
- Li, C., Chen, C., Gao, L., Yang, S., Nguyen, V., Shi, X., Siminovitch, K., Kohalmi, S.E., Huang, S., Wu, K., Chen, X., and Cui, Y. (2015). The Arabidopsis SWI2/SNF2 chromatin remodeler BRAHMA regulates polycomb function during vegetative development and directly activates the flowering repressor gene *SVP*. *PLoS Genet* **11**, e1004944.
- Wang, F.X., Shang, G.D., Wu, L.Y., Xu, Z.G., Zhao, X.Y., and Wang, J.W. (2020). Chromatin Accessibility Dynamics and a Hierarchical Transcriptional Regulatory Network Structure for Plant Somatic Embryogenesis. *Dev Cell* **54**, 742-757 e748.
- Wu, M.F., Sang, Y., Bezhani, S., Yamaguchi, N., Han, S.K., Li, Z., Su, Y., Slewinski, T.L., and Wagner, D. (2012). SWI2/SNF2 chromatin remodeling ATPases overcome polycomb repression and control floral organ identity with the LEAFY and SEPALLATA3 transcription factors. *Proc Natl Acad Sci U S A* **109**, 3576-3581.
- Xu, M., Hu, T., Smith, M.R., and Poethig, R.S. (2016a). Epigenetic Regulation of Vegetative Phase Change in Arabidopsis. *Plant Cell* **28**, 28-41.
- Xu, M., Hu, T., Zhao, J., Park, M.Y., Earley, K.W., Wu, G., Yang, L., and Poethig, R.S. (2016b). Developmental Functions of miR156-Regulated SQUAMOSA PROMOTER BINDING PROTEIN-LIKE (SPL) Genes in Arabidopsis thaliana. *PLoS Genet* **12**, e1006263.

REVIEWER COMMENTS

Reviewer #1 (Remarks to the Author):

All points raised by me have been now addressed. I noticed a small problem with the iProX data description as it still refers to Maxquant with the authors say they did not use for analysis...

Reviewer #2 (Remarks to the Author):

The authors have addressed my questions. Nice work.

Reviewer #3 (Remarks to the Author):

There is a mistake LC-MS methods in the line 773: the loading flow rate should be probably 10 μ l/min not 10ml/min.

How mass spectrometer raw files were processed to obtain peaklist files needed for search in Protein Prospector? Were any algorithms combining spectra from repeated fragmentation of one peptide ion used, etc?

The names of raw mass spectrometric files (BRIP3/4-GFP, BRIP4/4-GFP-brm-1) do not reflect protein names in the manuscript (BCL7A/B). I could not find any Arabidopsis protein named BRIP3/4 therefore I assume that BRIP3=BCL7A and BRIP4=BCL7B. The names of files should be corrected.

The name BCL7A/B used in manuscript is debatable. These proteins were first described by Jorge Hernández-García and colleagues in 2022 and named BCL-domain homolog 1 (BDH1) and BDH2 (Hernández-García, J. et al. Comprehensive identification of SWI/SNF complex subunits underpins deep eukaryotic ancestry and reveals new plant components. *Commun Biol* 5, 549 (2022).). The naming convention is a minor issue but the article should be cited in the manuscript.

Raw mass spectrometric data provided by authors contain different number of biological replicates than described in the manuscript. There are 4 biological replicates of BRIP3-GFP (=BCL7A) and BRIP4-GFP (=BCL7B). There are two biological replicates described in the manuscript. It is difficult to deduce which replicates were used.

There are 3 control files in the repository but two of these files are identical. The duplicated file should be removed.

There is no list of all proteins identified therefore I processed the raw data and performed protein identification. Surprisingly I found SWI2D and SWI3A proteins – known components of SWI/SNF complex with very high scores and peptide numbers.

There are SWI3D (17 – 26 peptides) and SWI3A (10 – 13 peptides) proteins identified in BCL7B-GFP (=BRIP4-GFP) with very high scores. SWI3D was identified on 4th place of the list. SWI3C protein described in the manuscript was identified by 6 – 11 peptides with two times less peptides than SWI3D.

Authors do not show or discuss that results. It looks like BCL7B is a part of SWI/SNF complex containing SWI3D more often than the complex containing SWI3C.

SWI3D identifies in BCL7B-GFP-brm1 samples with less peptides (5 – 8) which is different than SWI3C which was not identified in brm-1. SWI3A identifies in brm mutant (samples BCL7B-GFP-brm-1) with scores similar to WT.

The situation is even more pronounced in BCL7A-GFP samples where SWI3D identifies with the highest number of peptides (20 – 22 peptides, first place on the list). It is much more than SWI3C which identifies with 1 – 5 peptides. Interestingly both SWI3D and SWI3A identify in brm mutant (BCL7A-GFP-brm-1) with peptide numbers comparable to WT.

Authors of the manuscript write that in absence of BRM, BCL7A/B is not able to immunoprecipitate SWI3C but do not notice that BRM is not needed to immunoprecipitate SWI3D and SWI3A. These results should not be ignored.

The other ATPase of chromatin remodeling complexes SYD has also been identified with high scores in most of samples (19-22 peptides in BCL7A-GFP/BCL7A-GFP-brm-1; 13-16 peptides in BCL7B-GFB and only 1 – 3 peptides in BCL7B-GFP-brm-1). MINU1, MINU2 ATPases are identified as well. These results are not shown in figures or tables (there are also no table of all identified proteins in the supplement). Authors only mention the identification of this proteins in discussion. It would be worth mentioning that SYD and MINU1 were identified with much higher peptide numbers than BRM in BCL7A-GFP samples but the situation is opposite in BCL7B where BRM is identified with higher numbers of peptides. It could indicate differences in affinity of BCL7A and BCL7B to different complexes.

The interaction of SYD, MINU1/2 with BCL7s corresponds to the observations described in lines 253 – 256 (“...Notably, the BCL7s binding sites that were unaffected in brm-1 were significantly enriched in both SYD and MINU binding sites, implying that the binding of BCLs at those sites may be directly regulated by SYD and/or MINU”). The fact of coimmunoprecipitation of these proteins could be mentioned also in the results.

Proteomic data presented in the manuscript are of good quality. The proteins of interest are identified with high numbers of peptides and scores. There are also not much of contaminant proteins. Two first replicates of the experiment (bio1 and bio2) of BCL7A/B-GFP were analysed on LC-MS two years earlier (2020) than the rest (analysed in 2022). I suppose that the data from 2022 were used for preparing tables and figures. Control data (IP1 and IP2) were obtained in 2022. I would prefer control samples run together with the rest of samples but this is a minor issue.

The data obtained during this experiments are valuable and could be useful in future (maybe for some meta-analyses) therefore I opt for correcting filenames and descriptions in the repository.

I attach tables with results of protein identification performed by authors with added results of my analyses (especially identification of proteins not shown in manuscript).

Supplementary Table 1. Summary of peptide scores of the BAS subunits identified by IP-MS in BCL7B-GFP bcl7b and BCL7B-GFP bcl7b brm-1. Added the results of identifications made with use of deposited RAW data.

		BCL7B-GFP				BRIP4-GFP = BCL7B-GFP* (from raw data)								BCL7B-GFP brm-1				BRIP4-GFP-brm1 = BCL7B-GFP-brm1* (from raw data)			
		IP1		IP2		IP1		IP2		IP3		IP4		IP1		IP2		IP1		IP2	
		Sum PEP Score	Un. PEP	Sum PEP Score	Un. PEP	Mascot Score	Un. PEP	Mascot Score	Un. PEP	Mascot Score	Un. PEP	Mascot Score	Un. PEP	Sum PEP Score	Un. PEP	Sum PEP Score	Un. PEP	Mascot Score	Un. PEP	Mascot Score	Un. PEP
AT5G55210	BCL7B(bait)	41,53	6	36,653	5	338	4	277	5	669	8	913	9	38,138	5	37,884	6	201	4	212	4
AT2G46020	BRM	61,532	19	45,688	12	1177	26	875	28	1348	33	1680	38	NA	0	NA	0	-		29	1
AT1G21700	SWI3C	66,289	16	61,259	13	383	6	219	6	457	12	555	11	NA	0	NA	0	-		-	
AT3G03460	BRIP1	3,695	1	6,71	2	-		-		115	3	98	3	NA	0	NA	0	-		-	
AT5G17510	BRIP2	39,933	8	43,91	8	214	4	216	5	312	6	337	7	NA	0	NA	0			-	
AT1G20670	BRD1	80,444	20	73,396	17	115	3	114	3	360	8	341	8	NA	0	NA	0	-		-	
AT1G76380	BRD2	27,943	10	18,546	8	-		102	3	28	1	58	2	NA	0	NA	0	-		-	
AT5G55040	BRD13	24,037	8	20,353	8	122	2	113	5	96	2	137	3	NA	0	NA	0	-		-	
AT5G14170	SWP73B	45,287	11	69,238	14	863	12	670	14	1258	17	826	14	74,15	16	59,464	14	251	8	404	7
AT1G18450	ARP4	95,354	20	149,017	22	1025	17	649	17	1331	19	1379	22	88,784	16	112,474	15	293	6	472	7
AT3G60830	ARP7	83,489	17	109,138	16	936	16	752	15	1692	17	1138	17	73,517	14	75,231	13	229	9	838	13
AT3G18780	ACT2	7,746	1	9,709	2	29	1	59	2	272	10	221	6	18,004	2	11,826	1	186	7	215	7
AT5G09810	ACT7	7,02	1	8,994	1	29	1	59	2	146	5	149	4	12,924	1	15,958	2	163	6	295	7
AT1G01160	GIF2	4,565	1	7,14	1	-		-		-		59	1	7,291	1	NA	0	-		-	
AT4G34430	SWI3D					877	17	810	26	969	24	1059	26					169	5	269	8
AT2G47620	SWI3A					655	13	399	10	588	13	772	14					269	6	536	12
AT2G28290	SYD					414.0	13	447.0	14	497.0	16	418.0	14					65.0	1	146.0	3
At3G06010	MINU1					496.0	12	532.0	15	412.0	13	907.0	15					29.0	1	28.0	1
At5g19310	MINU2					104.0	3	273.0	9	222.0	7	265.0	3							28.0	1

*RAW data processed with Mascot Distiller, proteins identified with Mascot working on TAIR10+GFP databas (mass tolerance 15ppm for MS and 20 ppm for MSMS, trypsin with 2 missed cleavage sites allowed), identification results processed and filtered by Mscan (mascot score ≥10).

		BCL7A-GFP				BRIP3-GFP = BCL7A-GFP* (from raw data)								BCL7A-GFP brm-1				BRIP3-GFP-brm1 = BCL7A-GFP-brm1* (from raw data)			
		IP1		IP2		IP1		IP2		IP3		IP4		IP1		IP2		IP1		IP2	
		Sum PEP Score	Un. PEP	Sum PEP Score	Un. PEP	Mascot Score	Un. PEP	Mascot Score	Un. PEP	Mascot Score	Un. PEP	Mascot Score	Un. PEP	Sum PEP Score	Un. PEP	Sum PEP Score	Un. PEP	Mascot Score	Un. PEP	Mascot Score	Un. PEP
AT4G22320	BCL7A(bait)	35,279	7	36,989	8	51	2	130	5	134	3	266	4	37,548	9	38,487	8	117	2	139	2
AT2G46020	BRM	7,368	2	9,296	5	115	3	405	12	309	8	485	14	NA	0	NA	0	-		-	
AT1G21700	SWI3C	4,837	2	4,49	2	74	2	27	1	263	5	100	3	NA	0	NA	0	-		-	
AT3G03460	BRIP1	NA	0	NA	0	-		-		43	1	-		NA	0	NA	0	-		-	
AT5G17510	BRIP2	11,451	5	10,639	4	-		-		85	3	119	4	NA	0	NA	0	-		-	
AT1G20670	BRD1	2,875	1	3,655	2	-		-		139	6	186	4	NA	0	NA	0	-		-	
AT1G76380	BRD2	NA	0	NA	0	-		-		34	1	-		NA	0	NA	0	-		-	
AT5G55040	BRD13	NA	0	NA	0	-		69	3	46	1	38	1	NA	0	NA	0	-		-	
AT5G14170	SWP73B	56,493	14	49,825	14	403	8	477	10	624	13	853	18	53,909	15	60,029	13	808	16	735	18
AT1G18450	ARP4	83,291	17	64,888	15	422	13	778	18	703	15	1422	17	62,928	14	83,312	15	711	13	761	16
AT3G60830	ARP7	67,6	12	42,543	13	572	13	544	12	1441	19	1545	23	57,629	16	76,086	11	1119	19	1068	15
AT3G18780	ACT2	NA	0	NA	0	32	1	29	1	216	4	165	7	6,337	1	3,373	1	102	4	61	1
AT5G09810	ACT7	NA	0	3,876	2	32	1	29	1	285	6	147	6	6,494	1	NA	0	102	4	61	1
AT1G01160	GIF2	NA	0	NA	0	-		-		-		-		NA	0	NA	0	-		-	
AT4G34430	SWI3D					763	22	820	22	903	20	1065	22					681	20	815	19
AT2G47620	SWI3A					332	11	477	13	965	18	1010	20					464	11	521	12
AT2G28290	SYD					493.0	16	549.0	18	414.0	11	416.0	11					245	12	329	10
At3G06010	MINU1					483.0	14	665.0	18	267.0	6	488.0	12					133.0	5	169.0	4
At5g19310	MINU2					87.0	3	343.0	10	42.0	1	172.0	5							27.0	1

*RAW data processed with Mascot Distiller, proteins identified with Mascot working on TAIR10+GFP databas (mass tolerance 15ppm for MS and 20 ppm for MSMS, trypsin with 2 missed cleavage sites allowed), identification results processed and filtered by Mscan (mascot score ≥10).

Response to the comments of Reviewers:

Reviewer #1 (Remarks to the Author):

All points raised by me have been now addressed. I noticed a small problem with the iProX data description as it still refers to Maxquant with the authors say they did not use for analysis...

Response: Thank you for pointing this out and we have updated the description.

Reviewer #2 (Remarks to the Author):

The authors have addressed my questions. Nice work.

Response: Many thanks to the reviewer for the support.

Reviewer #3 (Remarks to the Author):

There is a mistake LC-MS methods in the line 773: the loading flow rate should be probably 10 μ l/min not 10ml/min.

Response: Thank you for pointing this out, and we have corrected it in the manuscript (line 796).

How mass spectrometer raw files were processed to obtain peaklist files needed for search in Protein Prospector? Were any algorithms combining spectra from repeated fragmentation of one peptide ion used, etc?

Response: Sorry for the confusion. RAW files were processed by Sequest HT algorithm (Eng et al., 1994) which is implemented in Thermo Proteome Discoverer (version: 2.4.1.15) software. Tryptic peptides with a minimum length of 6 amino acids, maximum length of 144 amino acids, and 2 maximum missed cleavage sites were searched with a precursor mass tolerance of 10 ppm and fragment mass tolerance of 0.02 Da. Proteins were then identified with Thermo Proteome Discoverer against the *Arabidopsis thaliana* TAIR10 plus GFP database, following manufactures' instructions with default settings. Spectra from repeated fragmentation of one peptide ion were not combined. We no longer use Protein Prospector. We have updated this information in

the MS text (lines 799-806). Thanks!

The names of raw mass spectrometric files (BRIP3/4-GFP, BRIP4/4-GFP-brm-1) do not reflect protein names in the manuscript (BCL7A/B). I could not find any Arabidopsis protein named BRIP3/4 therefore I assume that BRIP3=BCL7A and BRIP4=BCL7B. The names of files should be corrected.

Response: Thank you for the comment. We have changed BRIP3 to BCL7A and BRIP4 to BCL7B.

The name BCL7A/B used in manuscript is debatable. These proteins were first described by Jorge Hernández-García and colleagues in 2022 and named BCL-domain homolog 1 (BDH1) and BDH2 (Hernández-García, J. et al. Comprehensive identification of SWI/SNF complex subunits underpins deep eukaryotic ancestry and reveals new plant components. *Commun Biol* 5, 549 (2022).). The naming convention is a minor issue but the article should be cited in the manuscript.

Response: Thank you for the point. We have added the citation (lines 134-135).

Raw mass spectrometric data provided by authors contain different number of biological replicates than described in the manuscript. There are 4 biological replicates of BRIP3-GFP (=BCL7A) and BRIP4-GFP (=BCL7B). There are two biological replicates described in the manuscript. It is difficult to deduce which replicates were used.

Response: Thank you for your comment, and we apologize for any confusion here. We used biological replicates 3 and 4 for BCL7A-GFP samples, and this information has been updated. For BCL7B-GFP IP, we noticed that BRM peptides were identified in the *BCL7B-GFP brm-1* samples. Given that BRM peptides in the *brm-1* background might result from contamination from *BCL7B-GFP* WT samples, we have repeated mass spectrometry experiments with BCL7B-GFP in both WT and *brm-1* backgrounds. In the new BCL7B-GFP IP under the *brm-1* mutant background, BRM peptides were not identified. The results confirmed the conclusion that BCL7B requires BRM for incorporation/assembly with BAS-specific subunits (SWI3C, BRIP2 and BRD1/2/13). We have updated mass spectrometry data for BCL7B. Thanks again!

There are 3 control files in the repository but two of these files are identical. The duplicated file should be removed.

Response: The duplicated raw files of WT samples were removed. Thanks again!

There is no list of all proteins identified therefore I processed the raw data and performed protein identification.

Response: Thank you for your comment. We have added the full list of proteins identified by mass spectrometry analysis in Supplementary Table 1.

Surprisingly I found SWI2D and SWI3A proteins – known components of SWI/SNF complex with very high scores and peptide numbers. There are SWI3D (17 – 26

peptides) and SWI3A (10 – 13 peptides) proteins identified in BCL7B-GFP (=BRIP4-GFP) with very high scores. SWI3D was identified on 4th place of the list. SWI3C protein described in the manuscript was identified by 6 – 11 peptides with two times less peptides than SWI3D. Authors do not show or discuss that results. It looks like BCL7B is a part of SWI/SNF complex containing SWI3D more often than the complex containing SWI3C.

SWI3D identifies in BCL7B-GFP-brm1 samples with less peptides (5 – 8) which is different than SWI3C which was not identified in brm-1. SWI3A identifies in brm mutant (samples BCL7B-GFP-brm-1) with scores similar to WT.

The situation is even more pronounced in BCL7A-GFP samples where SWI3D identifies with the highest number of peptides (20 – 22 peptides, first place on the list). It is much more than SWI3C which identifies with 1 – 5 peptides. Interestingly both SWI3D and SWI3A identify in brm mutant (BCL7A-GFP-brm-1) with peptide numbers comparable to WT.

Authors of the manuscript write that in absence of BRM, BCL7A/B is not able to immunoprecipitate SWI3C but do not notice that BRM is not needed to immunoprecipitate SWI3D and SWI3A. These results should not be ignored.

Response: Thank you for your feedback. We have presented these results in our revised manuscript as follow (lines 239-249):

Notably, SWI3A and SWI3D were immunoprecipitated by BCL7A or BCL7B in the *brm-1* mutant backgrounds with peptide numbers comparable to WT background. Thus, BRM is not necessary for BCL7A/B to immunoprecipitate SWI3D and SWI3A, in line with previous reports showing that BRM does not form complexes with SWI3D and SWI3A.

Interestingly, we observed a higher number of SWI3D peptides immunoprecipitated by BCL7B-GFP compared to SWIC (Fig. 2a and b). A similar result was also observed in BCL7A-GFP samples (Extended Data Fig. 5a-c and Supplementary Table 1). These results imply the possibility that BCL7A and BCL7B are more frequently associated with SWI/SNF complexes containing SWI3D than those containing SWI3C.

The other ATPase of chromatin remodeling complexes SYD has also been identified with high scores in most of samples (19-22 peptides in BCL7A-GFP/BCL7A-GFP-brm-1; 13-16 peptides in BCL7B-GFB and only 1 – 3 peptides in BCL7B-GFP-brm-1). MINU1, MINU2 ATPases are identified as well. These results are not shown in figures or tables (there are also no table of all identified proteins in the supplement). Authors only mention the identification of this proteins in discussion. It would be worth mentioning that SYD and MINU1 were identified with much higher peptide numbers than BRM in BCL7A-GFP samples but the situation is opposite in BCL7B where BRM is identified with higher numbers of peptides. It could indicate differences in affinity of BCL7A and BCL7B to different complexes.

Response: Thank you for your comment. We have presented these results in the manuscript as follow (lines 249-256):

SYD and MINU1/2 ATPases were identified in BCL7A/B-GFP samples (Fig. 2a-

c and Extended Data Fig. 5a-c). Notably, SYD showed relatively higher peptide numbers compared to BRM in BCL7A-GFP samples (Extended Data Fig. 5a-c). However, the scenario reversed in BCL7B-GFP plants, where BRM was identified with a higher number of peptides compared to SYD and MINU1/2 (Fig. 2a-c). This observation suggests that BCL7A might prefer to interact with SYD, whereas BCL7B exhibits a higher affinity for BRM ATPase compared to SYD and MINU1/2.

The interaction of SYD, MINU1/2 with BCL7s corresponds to the observations described in lines 253 – 256 (“...Notably, the BCL7s binding sites that were unaffected in *brm-1* were significantly enriched in both SYD and MINU binding sites, implying that the binding of BCLs at those sites may be directly regulated by SYD and/or MINU”). The fact of coimmunoprecipitation of these proteins could be mentioned also in the results.

Response: Thank you for the comment. We have added the wording in the MS (lines 274-276).

Proteomic data presented in the manuscript are of good quality. The proteins of interest are identified with high numbers of peptides and scores. There are also not much of contaminant proteins. Two first replicates of the experiment (bio1 and bio2) of BCL7A/B-GFP were analysed on LC-MS two years earlier (2020) than the rest (analysed in 2022). I suppose that the data from 2022 were used for preparing tables and figures. Control data (IP1 and IP2) were obtained in 2022. I would prefer control samples run together with the rest of samples but this is a minor issue.

Response: Thank you for pointing this out. For BCL7A-GFP samples, we used biological replicates 3 and 4. The mass spectrometry experiments with BCL7B-GFP in both WT and *brm-1* backgrounds have been repeated. Our lab routinely conducts control experiments using wild-type samples and does not detect subunits of the SWI/SNF complex in WT samples. Thanks!

The data obtained during this experiments are valuable and could be useful in future (maybe for some meta-analyses) therefore I opt for correcting filenames and descriptions in the repository.

Response: We sincerely appreciate your support of our work. We have made the necessary changes to the filenames and descriptions in the repository.

I attach tables with results of protein identification performed by authors with added results of my analyses (especially identification of proteins not shown in manuscript).

Response: Thank you for your careful evaluation. We have updated the table S1.

References

Eng, J.K., McCormack, A.L., and Yates, J.R. (1994). An approach to correlate tandem mass spectral data of peptides with amino acid sequences in a protein database. *J Am Soc Mass Spectrom* **5**, 976-989.

REVIEWERS' COMMENTS

Reviewer #3 (Remarks to the Author):

The authors have addressed my questions.

I could not see the link to LC-MS raw data in the last version of the manuscript. It should be added before publication.

Nice article.

Response to the comments of Reviewers:

Reviewer #3 (Remarks to the Author):

The authors have addressed my questions.

I could not see the link to LC-MS raw data in the last version of the manuscript. It should be added before publication.

Nice article.

Response: Thanks for the comments. The mass spectrometry proteomics data have been deposited in the iProX database under the dataset identifier IPX0005203001 [<https://www.iprox.cn//page/subproject.html?id=IPX0005203001>].